# EQUIVARIANT AND STABLE POSITIONAL ENCODING FOR MORE POWERFUL GRAPH NEURAL NETWORKS

**Haorui Wang**[1]*, **Haoteng Yin**[2], **Muhan Zhang**[3,4], **Pan Li**[2]

[1]School of Computer Science, Wuhan University
[2]Department of Computer Science, Purdue University
[3]Institute for Artificial Intelligence, Peking University, & [4]BIGAI
hr_wang@whu.edu.cn,muhan@pku.edu.cn,{yinht,panli}@purdue.edu

## ABSTRACT

Graph neural networks (GNN) have shown great advantages in many graph-based learning tasks but often fail to predict accurately for a task based on sets of nodes such as link/motif prediction and so on. Many works have recently proposed to address this problem by using random node features or node distance features. However, they suffer from either slow convergence, inaccurate prediction or high complexity. In this work, we revisit GNNs that allow using positional features of nodes given by positional encoding (PE) techniques such as Laplacian Eigenmap, Deepwalk, etc.. GNNs with PE often get criticized because they are not generalizable to unseen graphs (inductive) or stable. Here, we study these issues in a principled way and propose a provable solution, a class of GNN layers termed PEG with rigorous mathematical analysis. PEG uses separate channels to update the original node features and positional features. PEG imposes permutation equivariance w.r.t. the original node features and rotation equivariance w.r.t. the positional features simultaneously. Extensive link prediction experiments over 8 real-world networks demonstrate the advantages of PEG in generalization and scalability.[1]

## 1 INTRODUCTION

Graph neural networks (GNN), inheriting from the power of neural networks (Hornik et al., 1989), have recently become the de facto standard for machine learning with graph-structured data (Scarselli et al., 2008). While GNNs can easily outperform traditional algorithms for single-node tasks (such as node classification) and whole graph tasks (such as graph classification), GNNs predicting over a set of nodes often achieve subpar performance. For example, for link prediction, GNN models, such as GCN (Kipf & Welling, 2017), GAE (Kipf & Welling, 2016) may perform even worse than some simple heuristics such as common neighbors and Adamic Adar (Liben-Nowell & Kleinberg, 2007) (see the performance comparison over the networks Collab and PPA in Open Graph Benchmark (OGB) (Hu et al., 2020)). Similar issues widely appear in node-set-based tasks such as network motif prediction (Liu et al., 2022; Besta et al., 2021), motif counting (Chen et al., 2020), relation prediction (Wang et al., 2021a; Teru et al., 2020) and temporal interaction prediction (Wang et al., 2021b), which posts a big concern for applying GNNs to these relevant real-world applications.

The above failure is essentially induced by the loss of node identities during the intrinsic computation of GNNs. Nodes that get matched under graph automorphism will be associated with the same representation by GNNs and thus are indistinguishable (see Fig. 1(i)). A naive way to solve this problem is to pair GNNs with one-hot encoding as the extra node feature. However, it violates the fundamental inductive bias, i.e., permutation equivariance which GNNs are designed for, and thus may lead to poor generalization capability: The obtained GNNs are not transferable (inductive) across different node sets and different graphs or stable to network perturbation.

Many works have been recently proposed to address such an issue. The key idea is to use augmented node features, where either random features (RF) or deterministic distance encoding (DE) can be adopted. Interested readers may refer to the book chapter (Li & Leskovec, 2021) for detailed discussion. Here we give a brief review. RF by nature distinguishes nodes and guarantees permutation equivariance if the distribution to generate RF keep invariant across the nodes. Although GNNs paired with RF have been proved to be more expressive (Murphy et al., 2019; Sato et al., 2021), the training procedure is often hard to converge and the prediction is noisy and inaccurate due to the

---

*Wang was an intern at Purdue University when doing this project.
[1]Code available at `https://github.com/Graph-COM/PEG`

Figure 1: Illustration of Positional Encoding (PE): (i) GNNs cannot distinguish nodes a,b,c,d because the graph has automorphism where $a$ is mapped to $b, c, d$. GNNs fail to predict whether (a,b) or (a,d) is more likely to have a link; (ii) PE associates each node with extra positional features that may distinguish nodes; (iii) An example of PE uses the eigenvectors that correspond to the 2nd and 3rd smallest eigenvalues of the graph Laplacian as positional features (denoted as the 2-dim vectors besides each node). The proposed GNN layer PEG keeps rotation equivariance when processing these features.

injected randomness (Abboud et al., 2020). On the other hand, DE defines extra features by using the distance from a node to the node set where the prediction is to be made (Li et al., 2020). This technique is theoretically sound and empirically performs well (Zhang & Chen, 2018; Li et al., 2020; Zhang et al., 2020). But it introduces huge memory and time consumption. This is because DE is specific to each node set sample and no intermediate computational results, e.g., node representations in the canonical GNN pipeline can be shared across different samples.

To alleviate the computational cost of DE, absolute positions of nodes in the graph may be used as the extra features. We call this technique as positional encoding (PE). PE may approximate DE by measuring the distance between positional features and can be shared across different node set samples. However, the fundamental challenge is how to guarantee the GNNs trained with PE keep permutation equivariant and stable. Using the idea of RF, previous works randomize PE to guarantee permutation equivariance. Specifically, You et al. (2019) designs PE as the distances from a node to some randomly selected anchor nodes. However, the approach suffers from slow convergence and achieves merely subpar performance. Srinivasan & Ribeiro (2020) states that PE using the eigenvectors of the randomly permuted graph Laplacian matrix keeps permutation equivariant. Dwivedi & Bresson (2020); Kreuzer et al. (2021) argue that such eigenvectors are unique up to their signs and thus propose PE that randomly perturbs the signs of those eigenvectors. Unfortunately, these methods may have risks. They cannot provide permutation equivariant GNNs when the matrix has multiple eigenvalues, which thus are dangerous when applying to many practical networks. For example, large social networks, when not connected, have multiple 0 eigenvalues; small molecule networks often have non-trivial automorphism that may give multiple eigenvalues. Even if the eigenvalues are distinct, these methods are unstable. We prove that the sensitivity of node representations to the graph perturbation depends on the inverse of the smallest gap between two consecutive eigenvalues, which could be actually large when two eigenvalues are close (Lemma 3.4).

In this work, we propose a principled way of using PE to build more powerful GNNs. *The key idea is to use separate channels to update the original node features and positional features. The GNN architecture keeps not only permutation equivariant w.r.t. node features but also rotation equivariant w.r.t. positional features.* This idea applies to a broad range of PE techniques that can be formulated as matrix factorization (Qiu et al., 2018) such as Laplacian Eigenmap (LE) (Belkin & Niyogi, 2003) and Deepwalk (Perozzi et al., 2014). We design a GNN layer PEG that satisfies such requirements. PEG is provably stable: In particular, we prove that the sensitivity of node representations learnt by PEG only depends on the gap between the $p$th and $(p+1)$th eigenvalues of the graph Laplacian if $p$-dim LE is adopted as PE, instead of the smallest gap between any two consecutive eigenvalues that previous works have achieved.

PEG gets evaluated in the most important node-set-based task, link prediction, over 8 real-world networks. PEG achieves comparable performance with strong baselines based on DE while having much lower training and inference complexity. PEG achieves significantly better performance than other baselines without using DE. Such performance gaps get enlarged when we conduct domain-shift link prediction, where the networks used for training and testing are from different domains, which effectively demonstrates the strong generalization and transferability of PEG.

## 1.1 OTHER RELATED WORKS

As long as GNNs can be explained by a node-feature-refinement procedure (Hamilton et al., 2017; Gilmer et al., 2017; Morris et al., 2019; Veličković et al., 2018; Klicpera et al., 2019; Chien et al., 2021), they suffer from the aforementioned node ambiguity issue. Some GNNs cannot be explained as node-feature refinement as they directly track node-set representations (Maron et al., 2019b; Morris

et al., 2019; Chen et al., 2019; Maron et al., 2019a). However, their complexity is high, as they need to compute the representation of each node set of certain size. Moreover, they were only studied for graph-level tasks. EGNN (Satorras et al., 2021) seems to be a relevant work as it studies when the nodes have physical coordinates given in prior. However, no analysis of PE has been provided.

A few works studied the stability of GNNs by using tools like graph scattering transform (for graph-level representation) (Gama et al., 2019a;b) and graph signal filtering (for node-level representations) (Levie et al., 2021; Ruiz et al., 2020; 2021; Gama et al., 2020; Nilsson & Bresson, 2020). They all focus on justifying the stability of the canonical GNN pipeline, graph convolution layers in particular. None of them consider positional features let alone the stability of GNNs using PE.

## 2  NOTATIONS AND PRELIMINARIES

In this section, we prepare the notations and preliminaries that are useful later. First, we define graphs.

**Definition 2.1** (Graph). Unless specified, we always consider undirected graphs of $N$ nodes and let $[N] = \{1, 2, ..., N\}$. One such graph can be denoted as $\mathcal{G} = (A, X)$, where $A$ is the adjacency matrix. $X \in \mathbb{R}^{N \times F}$ denotes the node features, where the $v$th row, $X_v$, is the feature vector of node $v$. A graph may have self loops, i.e., $A$ has nonzero diagonals. Denote $D$ as the diagonal degree matrix where $D_{vv} = \sum_{u \in [N]} A_{vu}$ for $v \in [N]$. Let $d_{\max} = \max_{v \in [N]} D_{vv}$. Denote the normalized adjacency matrix as $\hat{A} = D^{-\frac{1}{2}} A D^{-\frac{1}{2}}$ and the normalized Laplacian matrix as $L = I - \hat{A}$.

**Definition 2.2** (Permutation). An $N$-dim permutation matrix $P$ is a matrix in $\{0, 1\}^{N \times N}$ where each row and each column has only one single 1. All such matrices are collected in $\Pi(N)$, simplified as $\Pi$.

We denote the vector $\ell_2$-norm as $\| \cdot \|$, the Frobenius norm as $\| \cdot \|_F$ and the operator norm as $\| \cdot \|_{\mathrm{op}}$.

**Definition 2.3** (Graph-matching). Given two graphs $\mathcal{G}^{(i)} = (A^{(i)}, X^{(i)})$ and their normalized Laplacian matrices $L^{(i)}$ for $i \in \{1, 2\}$, their matching can be denoted by a permutation matrix $P \in \Pi$ that best aligns the graph structures and the node features.
$$P^*(\mathcal{G}^{(1)}, \mathcal{G}^{(2)}) \triangleq \arg\min_{P \in \Pi} \|L^{(1)} - P L^{(2)} P^T\|_F + \|X^{(1)} - P X^{(2)}\|_F$$
Using $L$ instead of $A$ to represent graph structures is for notational simplicity. Actually for an unweighted graph, there is a bijective mapping between $L$ and $A$. One can rewrite the first term with $A$. Later, we use $P^*$ by not specifying $\mathcal{G}_1$ and $\mathcal{G}_2$ if there is no confusion. The distance between the two graphs can be defined as $d(\mathcal{G}_1, \mathcal{G}_2) = \|L^{(1)} - P^* L^{(2)} P^{*T}\|_F + \|X^{(1)} - P^* X^{(2)}\|_F$.

Next, we review eigenvalue decomposition and summarize arguments on its uniqueness in Lemma 2.6.

**Definition 2.4** (Eigenvalue Decomposition (EVD)). For a positive semidefinite (PSD) matrix $B \in \mathbb{R}^{N \times N}$, it has eigenvalue decomposition $B = U \Lambda U^T$ where $\Lambda$ is a real diagonal matrix with the eigenvalues of $B$, $0 \le \lambda_1 \le \lambda_2 \le ... \le \lambda_N$ as its diagonal components. $U = [u_1, u_2, ..., u_N]$ is an orthogonal matrix where $u_i \in \mathbb{R}^N$ is the $i$th eigenvector, i.e. $B u_i = \lambda_i u_i$.

**Definition 2.5** (Orthogonal Group in the Euclidean space). $\mathrm{SO}(k) = \{Q \in \mathbb{R}^{k \times k} | Q Q^T = Q^T Q = I\}$ includes all $k$-by-$k$ orthogonal matrices. A subgroup of $\mathrm{SO}(k)$ includes all diagonal matrices with $\pm 1$ as the diagonal components, $\mathrm{SN}(k) = \{Q \in \mathbb{R}^{k \times k} | Q_{ii} \in \{-1, 1\}, Q_{ij} = 0, \text{ for } i \ne j\}$.

**Lemma 2.6.** EVD is not unique. If all the eigenvalues are distinct, i.e., $\lambda_i \ne \lambda_j$, $U$ is unique up to the signs of its columns, i.e., replacing $u_i$ by $-u_i$ also gives EVD. If there are multiple eigenvalues, say $(\lambda_{i-1} <) \lambda_i = \lambda_{i+1} = ... = \lambda_{i+k-1} (< \lambda_{i+k})$, then $[u_i, u_{i+1}, ..., u_{i+k-1}]$ lie in an orbit induced by the orthogonal group $\mathrm{SO}(k)$, i.e., replacing $[u_i, u_{i+1}, ..., u_{i+k-1}]$ by $[u_i, u_{i+1}, ..., u_{i+k-1}] Q$ for any $Q \in \mathrm{SO}(k)$ while keeping eigenvalues and other eigenvectors unchanged also gives EVD.

Next, we define Positional Encoding (PE), which associates each node with a vector in a metric space where the distance between two vectors can represent the distance between the nodes in the graph.

**Definition 2.7** (Positional Encoding). Given a graph $\mathcal{G} = (A, X)$, PE works on $A$ and gives $Z = \mathrm{PE}(A) \in \mathbb{R}^{N \times p}$ where each row $Z_v$ gives the positional feature of node $v$.

The absolute values given by PE may not be useful while the distances between the positional features are more relevant. So, we define PE-matching that allows rotation to best match positional features, which further defines the distance between two collections of positional features.

**Definition 2.8** (PE-matching). Consider two groups of positional features $Z^{(1)}, Z^{(2)} \in \mathbb{R}^{N \times p}$. Their matching is given by $Q^*(Z^{(1)}, Z^{(2)}) \triangleq \arg\min_{Q \in \mathrm{SO}(p)} \|Z^{(1)} - Z^{(2)} Q\|_F$. Later, $Q^*$ is used if it causes no confusion. Define the distance between them as $\eta(Z^{(1)}, Z^{(2)}) = \|Z^{(1)} - Z^{(2)} Q^*\|_F$.

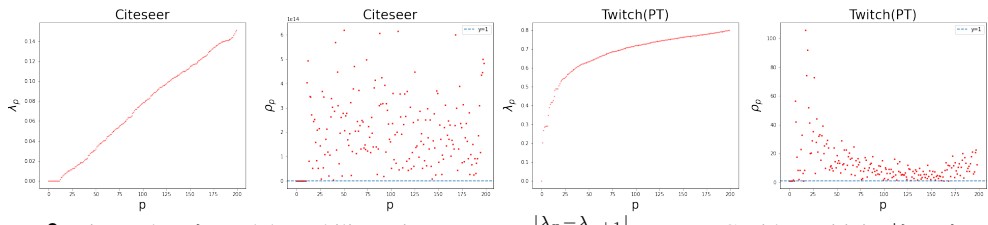

Figure 2: Eigenvalues $\lambda_p$ and the stability ratio $\rho_p = \frac{|\lambda_p - \lambda_{p+1}|}{\min_{1 \leq k \leq p} |\lambda_k - \lambda_{k+1}|}$. PEG with sensitivity $|\lambda_p - \lambda_{p+1}|^{-1}$ is far more stable than previous methods with sensitivity $\max_{1 \leq k \leq p} |\lambda_k - \lambda_{k+1}|^{-1}$. Over Citeseer, $\rho_p$ is extremely large because there are multiple eigenvalues ($\rho_p$ is still finite due to the numerical approximation of the eigenvalues). Over Twitch(PT), even if there are no multiple eigenvalues, $\rho_p$ is mostly larger than 10.

## 3 EQUIVARIANT AND STABLE POSITIONAL ENCODING FOR GNN

In this section, we will study the equivariance and stability of a GNN layer using PE .

### 3.1 OUR GOAL: BUILDING PERMUTATION EQUIVARIANT AND STABLE GNN LAYERS

A GNN layer is expected to be permutation equivariant and stable. These two properties, if satisfied, guarantee that the GNN layer is transferable and has better generalization performance. Permutation equivariance implies that model predictions should be irrelevant to how one indexes the nodes, which captures the fundamental inductive bias of many graph learning problems: A permutation equivariant layer can make the same prediction over a new testing graph as that over a training graph if the two graphs match each other perfectly, i.e. the distance between them is 0. Stability is actually an even stronger condition than permutation equivariance because it characterizes how much gap between the predictions of the two graphs is expected when they do not perfectly match each other.

Specifically, given a graph $A$ with node features $X$, we consider a GNN layer $\tilde{g}$ updating the node features, denoted as $\hat{X} = \tilde{g}(A, X)$. We define permutation equivariance and stability as follows.

**Definition 3.1** (Permutation Equivariance). A GNN layer $\tilde{g}$ is permutation equivariant, if for any $P \in \Pi$ and any graph $\mathcal{G} = (A, X)$, $P\tilde{g}(A, X) = \tilde{g}(PAP^T, PX)$.

**Definition 3.2** (Stability). A GNN layer $\tilde{g}$ is claimed to be stable, if there exists a constant $C > 0$, for any two graphs $\mathcal{G}^{(1)} = (A^{(1)}, X^{(1)})$ and $\mathcal{G}^{(2)} = (A^{(2)}, X^{(2)})$, letting $P^* = P^*(\mathcal{G}^{(1)}, \mathcal{G}^{(2)})$ denote their matching, $\tilde{g}$ satisfies $\|\tilde{g}(A^{(1)}, X^{(1)}) - P^*\tilde{g}(A^{(2)}, X^{(2)})\|_F \leq Cd(\mathcal{G}^{(1)}, \mathcal{G}^{(2)})$. By setting $A^{(1)} = PA^{(2)}P^T$ and $X^{(1)} = PX^{(2)}$ for some $P \in \Pi$, the RHS becomes zero, so a stable $\tilde{g}$ makes the LHS zero too. So, stability is a stronger condition than permutation equivariance.

Our goal is to guarantee that the GNN layer that utilizes PE is permutation equivariant and stable. To distinguish from the GNN layer $\tilde{g}$ that does not use PE, we use $g$ to denote a GNN layer that uses PE, which takes the positional features $Z$ as one input and may update both node features $X$ and $Z$, i.e., $(\hat{X}, \hat{Z}) = g(A, X, Z)$. Now, we may define PE-equivariance and PE-stability for the GNN layer $g$.

**Definition 3.3** (PE-stability & PE-equivariance). Consider a GNN layer $g$ that uses PE. When it works on any two graphs $\mathcal{G}^{(i)} = (A^{(i)}, X^{(i)})$, $i \in \{1, 2\}$ and gives $(\hat{X}^{(i)}, \hat{Z}^{(i)}) = g(A^{(i)}, X^{(i)}, Z^{(i)})$, let $P^*$ be the matching between the two graphs. $g$ is PE-stable, if for some constant $C > 0$ we have

$$\|\hat{X}^{(1)} - P^*\hat{X}^{(2)}\|_F + \eta(\hat{Z}^{(1)}, P^*\hat{Z}^{(2)}) \leq Cd(\mathcal{G}^{(1)}, \mathcal{G}^{(2)}). \tag{1}$$

Recall $\eta(\cdot, \cdot)$ measures the distance between two sets of positional features as defined in Def. 2.8. Similar as above, a weaker condition of PE-stability is PE-equivariance: If $A^{(1)} = PA^{(2)}P^T$ and $X^{(1)} = PX^{(2)}$ for some $P \in \Pi$, we expect a perfect match between the updated node features and positional features, $\hat{X}^{(1)} = P\hat{X}^{(2)}$ and $\eta(\hat{Z}^{(1)}, P\hat{Z}^{(2)}) = 0$.

Note that previous works also consider $g$ that updates only node features, i.e., $\hat{X} = g(A, X, Z)$. In this case, PE-stability can be measured by removing the second term on $\hat{Z}$ from Eq.1.

### 3.2 PE-STABLE GNN LAYERS BASED ON LAPLACIAN EIGENMAP AS POSITIONAL ENCODING

To study the requirement of the GNN layer that achieves PE-stability, let us start from a particular PE technique, i.e., Laplacian eigenmap (LE) (Belkin & Niyogi, 2003) to show the key insight behind. Later, we generalize the concept to some other PE. Let $Z_{LE}$ denote LE, which includes the eigenvectors that correspond to the $p$ smallest eigenvalues of the normalized Laplacian matrix $L$.

**Previous works failed to design PE-equivariant or PE-stable GNN layers.** Srinivasan & Ribeiro (2020) claims that if a GNN layer $\tilde{g}(A, X)$ that does not rely on PE is permutation equivariant, PE-equivariance may be kept by adding $Z_{LE}$ to node features, i.e., $g(A, X, Z_{LE}) = \tilde{g}(A, X + \text{MLP}(Z_{LE}))$, where $\text{MLP}(\cdot)$ is a multilayer perceptron that adjusts the dimension properly. This statement is problematic when $Z_{LE}$ is not unique given the graph structure $A$. Specifically, though sharing graph structure $A^{(1)} = A^{(2)}$, if different implementations lead to different LEs $Z_{LE}^{(1)} \neq Z_{LE}^{(2)}$, then $\tilde{g}(A^{(1)}, X^{(1)} + \text{MLP}(Z_{LE}^{(1)})) \neq \tilde{g}(A^{(2)}, X^{(2)} + \text{MLP}(Z_{LE}^{(2)}))$, which violates PE-equivariance. Srinivasan & Ribeiro (2020) suggests using graph permutation augmentation to address the issue, which makes assumptions on an invariant distribution of $Z_{LE}$ that may not be guaranteed empirically.

Dwivedi & Bresson (2020); Kreuzer et al. (2021) claim the uniqueness of $Z_{LE}$ up to the signs and suggest building a GNN layer that uses random $Z_{LE}$ as $g(A, X, Z_{LE}) = \tilde{g}(A, X + \text{MLP}(Z_{LE}S))$ where $S$ is uniformly at random sampled from $\text{SN}(p)$. They expect PE-equivariance in the sense of expectation. However, this statement is generally incorrect because it depends on the condition that *all eigenvalues have to be distinct* as stated in Lemma 2.6. Actually, for multiple eigenvalues, there are infinitely many eigenvectors that lie in the orbit induced by the orthogonal group. Although many real graphs have multiple eigenvalues such as disconnected graphs or graphs with some non-trivial automorphism, one may argue that the methods work when the eigenvalues are all distinct. However, the above failure may further yield PE-instability even when the eigenvalues are distinct but have small gaps due to the following lemma.

**Lemma 3.4.** For any PSD matrix $B \in \mathbb{R}^{N \times N}$ without multiple eigenvalues, set positional encoding $\text{PE}(B)$ as the eigenvectors given by the smallest $p$ eigenvalues sorted as $0 = \lambda_1 < \lambda_2 < ... < \lambda_p(< \lambda_{p+1})$ of $B$. For any sufficiently small $\epsilon > 0$, there exists a perturbation $\triangle B$, $\|\triangle B\|_F \leq \epsilon$ such that

$$\min_{S \in \text{SN}(p)} \|\text{PE}(B) - \text{PE}(B + \triangle B)S\|_F \geq 0.99 \max_{1 \leq i \leq p} |\lambda_{i+1} - \lambda_i|^{-1} \|\triangle B\|_F + o(\epsilon). \quad (2)$$

Lemma 3.4 implies that small perturbation of graph structures may yield a big change of eigenvectors if there is a small eigengap. Consider two graphs $\mathcal{G}^{(1)} = (A^{(1)}, X^{(1)})$ and $\mathcal{G}^{(2)} = (A^{(2)}, X^{(2)})$, where $X^{(1)} = X^{(2)}$ and $A^{(2)}$ is $A^{(1)}$ with a small perturbation $\triangle A^{(1)}$. The perturbation is small enough so that the matching $P^*(\mathcal{G}^{(1)}, \mathcal{G}^{(2)}) = I$. However, the change of PE even after removing the effect of changing signs $\min_{S \in \text{SN}(p)} \|\text{PE}(A^{(1)}) - \text{PE}(A^{(2)})S\|$ could be dominated by the largest inverse eigengap among the first $p + 1$ eigenvalues $\max_{1 \leq i \leq p} |\lambda_{i+1} - \lambda_i|^{-1}$. In practice, it is hard to guarantee all these $p$ eigenpairs have large gaps, especially when a large $p$ is used to locate each node more accurately. Plugging this PE into a GNN layer gives the updated node features $\hat{X}^{(i)}$ a large change $\|\hat{X}^{(1)} - \hat{X}^{(2)}\|_F$ and thus violates PE-stability.

**PE-stable GNN layers.** Although a particular eigenvector may be not stable, the eigenspace, i.e., the space spanned by the columns of $\text{PE}(A)$ could be much more stable. This motivates our following design of the *PE-stable GNN layers*. Formally, we use the following lemma that can characterize the distance between the eigenspaces spanned by LEs of two graph Laplacians. The error is controlled by the inverse eigengap between the $p$th and $(p + 1)$th eigenvalues $\min_{i=1,2}\{|\lambda_p^{(i)} - \lambda_{p+1}^{(i)}|^{-1}\}$, which by properly setting $p$ is typically much smaller than $\max_{1 \leq k \leq p} |\lambda_k - \lambda_{k+1}|^{-1}$ in Lemma 3.4. We compute the ratio between these two values over some real-world graphs as shown in Fig. 2.

**Lemma 3.5.** For two PSD matrices $B^{(1)}, B^{(2)} \in \mathbb{R}^{N \times N}$, set $\text{PE}(B)$ as the eigenvectors given by the $p$ smallest eigenvalues of $B$. Suppose $B^{(i)}$ has eigenvalues $0 = \lambda_1^{(i)} \leq \lambda_2^{(i)} \leq ... \leq \lambda_p^{(i)} \leq \lambda_{p+1}^{(i)}$ and $\delta = \min_{i=1,2}\{|\lambda_p^{(i)} - \lambda_{p+1}^{(i)}|^{-1}\} < \infty$. Then, for any permutation matrix $P \in \Pi$,

$$\eta(\text{PE}(B^{(1)}), P\text{PE}(B^{(2)})) \leq 2^{\frac{3}{2}} \delta \min\{\sqrt{p}\|B^{(1)} - PB^{(2)}P^T\|_{\text{op}}, \|B^{(1)} - PB^{(2)}P^T\|_F\} \quad (3)$$

Inspired by the stability of the eigenspace, the idea to achieve PE-stability is to make the GNN layer invariant to the selection of bases of the eigenspace for the positional features. So, our proposed PE-stable GNN layer $g$ that uses PE should satisfy two necessary conditions: 1) Permutation equivariance w.r.t. all features; 2) Rotation equivariance w.r.t. positional features, i.e.,

$$\textbf{PE-stable layer cond. 1:} \qquad (P\hat{X}, P\hat{Z}) = g(PAP^T, PX, PZ), \ \forall P \in \Pi(N), \quad (4)$$

$$\textbf{PE-stable layer cond. 2:} \qquad (\hat{X}, \hat{Z}Q) = g(A, X, ZQ), \ \forall Q \in \text{SO}(p). \quad (5)$$

Rotation equivariance reflects the eigenspace instead of a particular selection of eigenvectors and thus achieves much better stability. Interestingly, these requirements can be satisfied by EGNN recently proposed (Satorras et al., 2021) as one can view the physical coordinates of objects considered by EGNN as the positional features. EGNN gets briefly reviewed in Appendix F. Thm. 3.6 proves PE-equivariance under the conditions Eqs. 4 and 5.

**Theorem 3.6.** A GNN layer $g(A, X, Z)$ that uses $Z = Z_{\text{LE}}$ and satisfies Eqs. 4,5 is PE-equivariant if the $p$th and $(p+1)$th eigenvalues of the normalized Laplacian matrix $L$ are different, i.e., $\lambda_p \neq \lambda_{p+1}$.

Note that satisfying Eqs. 4,5 is insufficient to guarantee PE-stability that depends on the form of $g$.

We implement $g$ in our model PEGN with further simplification which has already achieved good empirical performance: Use a GCN layer with edge weights according to the distance between the end nodes of the edge and keep the positional features unchanged. This gives **the PEG layer**,

$$\textbf{PEG:} \quad g_{\text{PEG}}(A, X, Z) = \left( \psi \left[ \left( \hat{A} \odot \Xi \right) XW \right], Z \right), \text{ where } \Xi_{uv} = \phi \left( \|Z_u - Z_v\| \right), \forall u, v \in [N]. \quad (6)$$

Here $\psi$ is an element-wise activation function, $\phi$ is an MLP mapping from $\mathbb{R} \to \mathbb{R}$ and $\odot$ is the Hadamard product. Note that if $\hat{A}$ is sparse, only $\Xi_{uv}$ for an edge $uv$ needs to be computed.

We may also prove that the PEG layer $g_{\text{PEG}}$ satisfies the even stronger condition PE-stability.

**Theorem 3.7.** Consider two graphs $\mathcal{G}^{(i)} = (A^{(i)}, X^{(i)})$, $i = 1, 2$. Denote their normalized Laplacian matrices' $p$th eigenvalue as $\lambda_p^{(i)}$ and $(p+1)$th eigenvalue as $\lambda_{p+1}^{(i)}$. Assume that $\delta = \min_{i=1,2}(\lambda_{p+1}^{(i)} - \lambda_p^{(i)})^{-1} < \infty$. Also, assume that $\psi$ and $\phi$ in the PEG layer in Eq. 6 are $\ell_\psi, \ell_\phi$-Lipschitz continuous respectively. Then, the PEG layer that uses $Z = Z_{LE}$ satisfies PE-stability with the constant $C$ in Eq. 1 as $C = [(7\delta \|X^{(1)}\|_{\text{op}} + 2d_{\max}^{(2)})\ell_\psi \ell_\phi \|W\|_{\text{op}} + 3\delta]$.

Note that to achieve PE-stability, we need to normalize the node initial features to keep $\|X^{(1)}\|_{\text{op}}$ bounded, and control $\|W\|_{\text{op}}$ and $\ell_\phi$. In practice $\ell_\psi \leq 1$ is typically satisfied, e.g. setting $\psi$ as ReLU. Here, the most important term is $\delta$. PE-stability may only be achieved when there is an eigengap between $\lambda_p$ and $\lambda_{p+1}$ and the larger eigengap, the more stable. This observation may be also instructive to select the $p$ in practice. As previous works may encounter a smaller eigengap (Lemma 3.4), their models will be generally more unstable.

Also, the simplified form of $g_{\text{PEG}}$ is *not necessary* to achieve PE-stability. However, as it has already given consistently better empirical performance than baselines, we choose to keep using $g_{\text{PEG}}$.

### 3.3 Generalized to Other PE techniques: DeepWalk and LINE

It is well known that LE, as to compute the smallest eigenvalues, can be written as a low-rank matrix optimization form $Z_{\text{LE}} \in \arg\min_{Z \in \mathbb{R}^{N \times p}} \text{tr}(L_N ZZ^T)$, s.t. $Z^T Z = I$. Other PE techniques, such as Deepwalk (Perozzi et al., 2014), Node2vec (Grover & Leskovec, 2016) and LINE (Tang et al., 2015) can be unified into a similar optimization form, where the positional features $Z$ are given by matrix factorization $M^* = Z'Z^T$ ($M^*$ may be asymmetric so $Z' \in \mathbb{R}^{N \times p}$ may not be $Z$) and $M^*$ satisfies

$$M^* = \arg\min_{M \in \mathbb{R}^{N \times N}} \ell_{\text{PE}}(A, M) \triangleq \text{tr}(f_+(A)g(M) + f_-(D)g(-M)), \quad \text{s.t.} \quad \text{rank}(M) \leq p \quad (7)$$

Here $f_+(\cdot) : \mathbb{R}^{N \times N} \to \mathbb{R}_{\geq 0}^{N \times N}$ typically revises $A$ by combining degree normalization and a power series of $A$, $f_-(\cdot) : \mathbb{R}^{N \times N} \to \mathbb{R}_{\geq 0}^{N \times N}$ corresponds to edge negative sampling that may be related to node degrees, and $g(\cdot)$ is a component-wise log-sigmoid function $x - \log(1 + \exp(x))$. E.g., in LINE, $f_+(A) = A$, $f_-(D) = c\mathbf{1}\mathbf{1}^T D^{\frac{3}{4}}$, for some positive constant $c$, where $\mathbf{1}$ is the all-one vector. More discussion on Eq.7 and the forms of $f_+$ and $f_-$ for other PE techniques are given in Appendix G.

According to the above optimization formulation of PE, all the PE techniques generate positional features $Z$ based on matrix factorization, thus are not unique. $Z$ always lies in the orbit induced by $\text{SO}(p)$, i.e., if $Z$ solves Eq.7, $ZQ$ for $Q \in \text{SO}(p)$ solves it too. A GNN layer $g$ still needs to satisfy Eqs. 4,5 to guarantee PE-equivariance. PE-stability actually asks even more.

**Theorem 3.8.** Consider a general PE($A$) technique that has an optimization objective as Eq.7, computes its optimal solution $M^*$ and decomposes $M^* = Z'Z^T$, s.t. $Z^T Z = I \in \mathbb{R}^{p \times p}$ to get positional features $Z$. Essentially, $Z$ consists of the right-singular vectors of $M^*$. If Eq.7 has a unique solution $M^*$, $\text{rank}(M^*) = p$ and $f_+, f_-$ therein satisfy $f_+(PAP^T) = Pf_+(A)P^T$ and $f_-(PDP^T) = Pf_-(D)P^T$, then a GNN layer $g(A, X, Z)$ that satisfies Eqs. 4,5 is PE-equivariant.

Note that the conditions on $f_+$ and $f_-$ are generally satisfied by Deepwalk, Node2vec and LINE. However, solving Eq.7 to get the optimal solution may not be guaranteed in practice because the low-rank constraint is non-convex. One may consider relaxing the low-rank constraint into the nuclear norm $\|M\|_* \leq \tau$ for some threshold $\tau$ (Recht et al., 2010), which reduces the optimization Eq.7 into a convex optimization and thus satisfies the conditions in Thm. 3.8. Empirically, this step and the step of computing SVD seem unnecessary according to our experiments. The PE-stability is related to the value of the smallest non-zero singular value of $M^*$. We leave the full characterization of PE-equivariance and PE-stability for the general PE techniques for the future study.

## 4 EXPERIMENTS

In this work, we use the most important node-set-based task link prediction to evaluate PEG, though it may apply to more general tasks. Two types of link prediction tasks are investigated: traditional link prediction (Task 1) and domain-shift link prediction (Task 2). In Task 1, the model gets trained, validated and tested over the same graphs while using different link sets. In Task 2, the graph used for training/validation is different from the one used for testing. Both tasks may reflect the effectiveness of a model while Task 2 may better demonstrate the model's generalization capability that strongly depends on permutation equivariance and stability. All the results are based on 10 times random tests.

### 4.1 THE EXPERIMENTAL PIPELINE OF PEG FOR LINK PREDICTION

We use PEG to build GNNs. The pipeline contains three steps. First, we adopt certain PE techniques to compute positional features $Z$. Second, we stack PEG layers according to Eq. 6. Suppose the final node representations are denoted by $(\hat{X}, Z)$. Third, for link prediction over $(u, v)$, we concatenate $(\hat{X}_u \hat{X}_v^T, Z_u Z_v^T)$ denoted as $H_{uv}$ and adopt MLP($H_{uv}$) to make final predictions. In the experiments, we test LE and Deepwalk as PE, and name the models PEG-LE and PEG-DW respectively. To verify the wide applicability of our theory, we also apply PEG to GraphSAGE Hamilton et al. (2017) by similarly using the distance between PEs to control the neighbor aggregation according to Eq. 6 and term the corresponding models PE-SAGE-LE and PE-SAGE-DW respectively.

For some graphs, especially those small ones where the union of link sets for training, validation and testing cover the entire graph, the model may overfit positional features and hold a large generalization gap. This is because the links used as labels to supervise the model training are also used to generate positional features. To avoid this issue, we consider a more elegant way to use the training links. We adopt a 10-fold partition of the training set. For each epoch, we periodically pick one fold of the links to supervise the model while using the rest links to compute positional features. Note that PEs for different folds can be pre-computed by removing every fold of links, which reduces computational overhead. In practice, the testing stage often corresponds to online service and has stricter time constraints. Such partition is not needed so there is no computational overhead for testing. We term the models trained in this way as PEG-LE+ and PEG-DW+ respectively.

### 4.2 TASK 1 — TRADITIONAL LINK PREDICTION

**Datasets.** We use eight real graphs, including Cora, CiteSeer and Pubmed (Sen et al., 2008), Twitch (RU), Twitch (PT) and Chameleon (Rozemberczki et al., 2021), DDI and COLLAB (Hu et al., 2021). Over the first 6 graphs, we utilize 85%, 5%, 10% to partition the link set that gives positive examples for training, validation and testing and pair them with the same numbers of negative examples (missing edges). For the two OGB graphs, we adopt the dataset splits in (Hu et al., 2020). The links for validation and test are removed during the training stage and the links for validation are also not used in the test stage. All the models are trained till the loss converges and the models with the best validation performance is used to test.

**Baselines.** We choose 6 baselines: *VGAE* (Kipf & Welling, 2016), *P-GNN* (You et al., 2019), *SEAL* (Zhang & Chen, 2018), *GNN Trans.* (Dwivedi & Bresson, 2020), *LE* (Belkin & Niyogi, 2003) and *Deepwalk (DW)* (Perozzi et al., 2014). VGAE is a variational version of GAE (Kipf & Welling, 2016) that utilizes GCN to encode both the structural information and node feature information, and then decode the node embeddings to reconstruct the graph adjacency matrix. SEAL is particularly designed for link prediction by using enclosing subgraph representations of target node-pairs. P-GNN randomly chooses some anchor nodes and aggregates only from these anchor nodes. GNN Trans. adopts LE as PE and merges LE into node features and utilizes attention mechanism to aggregate information from neighbor nodes. For VGAE, P-GNN and GNN Trans., the inner product of two node embeddings is adopted to represent links. LE and DW are two network embedding methods, where the obtained node embeddings are already positional features and directly used to predict links.

Table 1: Performance on the traditional link prediction tasks, measured in ROC AUC (mean±std%).

| Method | Feature | Cora | Citeseer | Pubmed | Twitch-RU | Twitch-PT | Chameleon |
|---|---|---|---|---|---|---|---|
| VGAE | N. | 89.89 ± 0.06 | 90.11 ± 0.08 | 94.62 ± 0.02 | 83.13 ± 0.07 | 82.89 ± 0.08 | 97.98 ± 0.01 |
|  | C. | 55.68 ± 0.05 | 61.45 ± 0.36 | 69.03 ± 0.03 | 85.37 ± 0.02 | 85.69 ± 0.09 | 83.13 ± 0.04 |
|  | O. | 83.97 ± 0.05 | 77.22 ± 0.04 | 82.54 ± 0.04 | 84.76 ± 0.09 | 87.91 ± 0.05 | 97.67 ± 0.04 |
|  | P. | 83.82 ± 0.12 | 78.68 ± 0.25 | 81.74 ± 0.15 | 85.06 ± 0.14 | 85.06 ± 0.14 | 97.91 ± 0.03 |
|  | R. | 68.43 ± 0.42 | 71.21 ± 0.78 | 69.31 ± 0.23 | 68.42 ± 0.43 | 68.49 ± 0.73 | 73.44 ± 0.53 |
|  | N. + P. | 87.96 ± 0.29 | 80.04 ± 0.60 | 85.26 ± 0.17 | 84.59 ± 0.37 | 88.27 ± 0.19 | 98.01 ± 0.12 |
| PGNN | N. + P. | 86.92 ± 0.02 | 90.26 ± 0.02 | 88.12 ± 0.06 | 83.21 ± 0.00 | 82.37 ± 0.02 | 94.25 ± 0.01 |
| GNN-Trans. | N. + P. | 79.31 ± 0.09 | 77.49 ± 0.02 | 81.23 ± 0.12 | 79.24 ± 0.33 | 75.44 ± 0.14 | 86.23 ± 0.12 |
| SEAL | N. + D. | 91.32 ± 0.91 | 89.49 ± 0.43 | 97.16 ± 0.28 | 92.12 ± 0.10 | **93.21 ± 0.06**$^\dagger$ | **99.31 ± 0.18**$^\dagger$ |
| LE | P. | 84.43 ± 0.02 | 78.36 ± 0.08 | 84.35 ± 0.04 | 78.80 ± 0.10 | 67.56 ± 0.02 | 88.47 ± 0.03 |
| DW | P. | 86.82 ± 0.18 | 87.93 ± 0.11 | 85.79 ± 0.06 | 83.10 ± 0.05 | 83.47 ± 0.03 | 92.15 ± 0.02 |
| PEG-DW | N. + P. | 89.51 ± 0.08 | 91.67 ± 0.12 | 87.68 ± 0.29 | 90.21 ± 0.04 | 89.67 ± 0.03 | 98.33 ± 0.01 |
| PEG-DW | C. + P. | 88.36 ± 0.10 | 88.48 ± 0.10 | 88.80 ± 0.11 | 90.32 ± 0.09 | 90.88 ± 0.05 | 97.30 ± 0.03 |
| PEG-LE | N. + P. | **94.20 ± 0.04**$^\dagger$ | 92.53 ± 0.09 | 87.70 ± 0.31 | 92.14 ± 0.05 | 92.28 ± 0.02 | 98.78 ± 0.02 |
| PEG-LE | C. + P. | 86.88 ± 0.03 | 76.96 ± 0.23 | 91.65 ± 0.02 | 90.21 ± 0.18 | 91.15 ± 0.13 | 98.73 ± 0.04 |
| PEG-DW+ | N. + P. | 93.32 ± 0.08 | 94.11 ± 0.14 | 97.88 ± 0.05 | 91.68 ± 0.01 | 92.15 ± 0.02 | 98.20 ± 0.01 |
| PEG-DW+ | C. + P. | 90.78 ± 0.09 | 91.22 ± 0.12 | 93.44 ± 0.05 | 90.22 ± 0.04 | 91.37 ± 0.05 | 97.50 ± 0.03 |
| PEG-LE+ | N. + P. | 93.78 ± 0.03 | **95.73 ± 0.09**$^\dagger$ | **97.92 ± 0.11**$^\dagger$ | 92.29 ± 0.11 | 92.37 ± 0.06 | 98.18 ± 0.02 |
| PEG-LE+ | C. + P. | 88.98 ± 0.14 | 78.61 ± 0.27 | 94.28 ± 0.05 | **92.35 ± 0.02**$^\dagger$ | 92.50 ± 0.06 | 97.79 ± 0.01 |

Table 2: Performance on OGB datasets, measured in Hit@20 and Hits@50 (mean±std%). Codes are run on CPU: Intel(R) Xeon(R) Gold 6248R @ 3.00GHz and GPU: NVIDIA QUADRO RTX 6000.

| Method | ogbl-ddi (Hits@20(%)) | | | | ogbl-collab (Hits@50(%)) | | | |
|---|---|---|---|---|---|---|---|---|
|  | training time | test time | Validation | test | training time | test time | Validation | test |
| GCN | 29min 27s | 0.20s | 55.27 ± 0.53 | 37.11 ± 0.21 | 1h38min17s | 1.38s | 52.71 ± 0.10 | 44.62 ± 0.01 |
| GraphSAGE | 14min 26s | 0.24s | 67.11 ± 1.21 | 52.81 ± 8.75 | 38min 10s | 0.83s | 57.16 ± 0.70 | 48.45 ± 0.80 |
| SEAL | 2h 04min 32s | 12.04s | 28.29 ± 0.38 | 30.23 ± 0.24 | 2h29min05s | 51.28s | 64.95 ± 0.04 | **54.71 ± 0.01**$^\dagger$ |
| PGNN | 9min 49.39s | 0.28s | 2.66 ± 0.16 | 1.74 ± 0.19 | N/A | N/A | N/A | N/A |
| GNN-trans. | 53min 26s | 0.35s | 15.63 ± 0.14 | 9.22 ± 0.21 | 1h52min22s | 1.86s | 18.17 ± 0.25 | 11.19 ± 0.42 |
| DW | 36min 41s | 0.23s | 0.04 ± 0.00 | 0.02 ± 0.00 | 34min40s | 1.08s | 53.64 ± 0.03 | 44.79 ± 0.02 |
| LE | 33min 42s | 0.29s | 0.09 ± 0.00 | 0.02 ± 0.00 | 37min22s | 1.23s | 0.10 ± 0.01 | 0.12 ± 0.02 |
| PEG-DW | 29min 56s | 0.27s | 56.47 ± 0.35 | 43.80 ± 0.32 | 1h42min 05s | 1.51s | 63.98 ± 0.05 | 54.33 ± 0.06 |
| PEG-LE | 30min 32s | 0.29s | 57.49 ± 0.47 | 30.16 ± 0.47 | 1h42min03s | 1.42s | 56.52 ± 0.12 | 48.76 ± 0.92 |
| PE-SAGE-DW | 25min 11s | 0.31s | 68.05 ± 0.96 | **56.16 ± 5.50**$^\dagger$ | 56min54s | 0.97s | 63.43 ± 0.48 | 54.17 ± 0.54 |
| PE-SAGE-LE | 26min 19s | 0.32s | 68.38 ± 0.78 | 51.49 ± 9.71 | 55min59s | 0.98s | 58.66 ± 0.55 | 49.75 ± 0.67 |
| PEG-DW+ | 48min 03s | 0.28s | 59.70 ± 6.87 | 47.93 ± 0.21 | 1h37min43s | 1.43s | 62.31 ± 0.19 | 53.71 ± 8.02 |
| PEG-LE+ | 51min 25.35s | 0.29s | 58.44 ± 1.71 | 28.32 ± 7.34 | 1h33min29s | 1.39s | 52.91 ± 1.24 | 45.96 ± 9.98 |

**Implementation details.** For VGAE, we consider six types of features: (1) node feature (N.): original feature of each node. (2) constant feature (C.): node degree. (3) positional feature (P.): PE extracted by Deepwalk. (4) one-hot feature (O.): one-hot encoding of node indices. (5) random feature (R.): random value $r_v \sim U(0, 1)$ (6) node feature and positional feature (N. + P.): concatenating the node feature and the positional feature. P-GNN uses node features and the distances from a node to some randomly selected anchor nodes as positional features (N. + P.). GNN Trans. utilizes node features and LE as positional features (N. + P.). SEAL adopt Double-Radius Node Labeling (DRNL) to compute deterministic distance features (N. + D.). For PEG, we consider node features plus positional features (N. + P.) or constant feature plus positional features (C. + P.).

**Results** are shown in Table 1 and Table 2. Over the small datasets in Table 1, VGAE with node features outperforms other features in Cora, Citeseer and Pubmed because the nodes features therein are mostly informative, while this is not true over the other three datasets. One-hot features and positional features almost achieve the same performance, which implies that that GNNs naively using PE makes positional features behave like one-hot features and may have instability issues. Constant features are not good because of the node ambiguity issues. Random features may introduce heavy noise that causes trouble in model convergence. Concatenating node features and positional features gives better performance than only using positional features but is sometimes worse than only using node features, which is again due to the instability issue by using positional features.

Although PGNN and GNN Trans. utilize positional features, they achieve subpar performance. SEAL outperforms all of the state-of-art methods, which again demonstrates that the effectiveness of distance features (Li et al., 2020; Zhang et al., 2020). PEG significantly outperforms all the baselines except SEAL. PEG+ by better using training sets achieves comparable or even better performance than SEAL, which demonstrates the contributions of the stable usage of PE. Moreover, PEG can achieve comparable performance in most cases even only paired with constant features, which benefits from the more expressive power given by PE (avoids node ambiguity). Note that PE without GNNs (LE or DW solo) does not perform well, which justifies the benefit by joining GNN with PE.

Table 3: Performance on the domain-shift link prediction tasks, measured in ROC AUC (mean±std%)

| Mehood | Features | Cora→Citeseer | Cora→Pubmed | ES→PT | EN→RU | PPI |
|---|---|---|---|---|---|---|
| | N. | 62.74 ± 0.03 | 63.53 ± 0.27 | 51.52 ± 0.17 | 60.08 ± 0.02 | 83.24 ± 0.20 |
| | C. | 62.16 ± 0.08 | 56.89 ± 0.36 | 82.72 ± 0.26 | 91.23 ± 0.07 | 75.27 ± 0.89 |
| VGAE | P. | 70.59 ± 0.03 | 79.83 ± 0.27 | 82.24 ± 0.24 | 81.42 ± 0.01 | 77.61 ± 0.47 |
| | R. | 68.44 ± 0.63 | 71.27 ± 0.37 | 71.26 ± 0.36 | 69.37 ± 0.35 | 75.88 ± 0.49 |
| | N. + P. | 76.45 ± 0.55 | 65.62 ± 0.42 | 71.46 ± 0.31 | 84.00 ± 0.28 | 84.67 ± 0.22 |
| PGNN | N. + P. | 85.02 ± 0.28 | 76.88 ± 0.42 | 70.41 ± 0.07 | 63.27 ± 0.27 | 80.84 ± 0.03 |
| GNN-Trans. | N. + P. | 61.60 ± 0.52 | 76.35 ± 0.17 | 63.44 ± 0.34 | 62.87 ± 0.22 | 79.82 ± 0.17 |
| SEAL | N. + D. | **91.36 ± 0.93**[†] | 89.62 ± 0.87 | **93.37 ± 0.05**[†] | 92.34 ± 0.14 | 88.99 ± 0.12 |
| LE | P. | 77.62 ± 0.04 | 84.03 ± 0.22 | 67.75 ± 0.09 | 77.57 ± 0.15 | 72.14 ± 0.82 |
| DW | P. | 86.48 ± 0.14 | 86.97 ± 0.06 | 83.56 ± 0.03 | 83.41 ± 0.04 | 85.18 ± 0.20 |
| PEG-DW | N. + P. | 89.91 ± 0.03 | 87.23 ± 0.34 | 91.82 ± 0.04 | 91.14 ± 0.02 | 87.36 ± 0.11 |
| PEG-DW | C. + P. | 89.75 ± 0.04 | 89.58 ± 0.08 | 91.27 ± 0.04 | 90.26 ± 0.07 | 86.42 ± 0.20 |
| PEG-LE | N. + P. | 82.57 ± 0.02 | 92.34 ± 0.28 | 91.61 ± 0.05 | 91.93 ± 0.13 | 85.34 ± 0.14 |
| PEG-LE | C. + P. | 79.60 ± 0.04 | 88.89 ± 0.13 | 91.38 ± 0.10 | 92.40 ± 0.10 | 85.22 ± 0.16 |
| PEG-DW+ | N. + P. | 91.15 ± 0.06 | 90.98 ± 0.03 | 91.24 ± 0.16 | 91.91 ± 0.02 | **89.92 ± 0.17**[†] |
| PEG-DW+ | C. + P. | 91.32 ± 0.01 | 90.93 ± 0.18 | 91.22 ± 0.02 | 92.14 ± 0.02 | 88.44 ± 0.29 |
| PEG-LE+ | N. + P. | 86.72 ± 0.05 | **93.34 ± 0.11**[†] | 91.67 ± 0.13 | 92.24 ± 0.19 | 86.77 ± 0.36 |
| PEG-LE+ | C. + P. | 87.62 ± 0.04 | 92.21 ± 0.20 | 91.37 ± 0.19 | **93.12 ± 0.21**[†] | 86.21 ± 0.27 |

Regarding the OGB datasets, PGNN and GNN Trans. do not perform well either. Besides, PGNN cannot scale to large graphs such as *collab*. The results of DW and LE demonstrate that the original positional features may only provide crude information, so pairing them with GNNs is helpful. PEG achieves the best results on *ddi*, and performs competitively with SEAL on *collab*. The complexity of PEG is comparable to canonical GNNs. Note that we do not count the time of PE as it relates to the particular implementation. If PEG is used for online serving when the time complexity is more important, PE can be computed in prior. For a fair comparison, we also do not count the pre-processing time of SEAL, PGNN or GNN Trans.. Most importantly, PEG performs significantly faster than SEAL on test because SEAL needs to compute distance features for every link while PE in PEG is shared by links. Interestingly, DW seems better than LE as a PE technique for large networks.

### 4.3 TASK 2 —DOMAIN-SHIFT LINK PREDICTION

**Datasets & Baselines.** Task 2 better evaluates the generalization capability of models. We consider 3 groups, including citation networks (cora→citeseer and cora→pubmed) (Sen et al., 2008), user-interaction networks (Twitch (EN)→Twitch (RU) and Twitch (ES)→Twitch (PT)) (Rozemberczki et al., 2021) and biological networks (PPI) (Hamilton et al., 2017). For citation networks and user-interaction networks, we utilize 95%, 5% dataset splitting for training and validation over the training graph, and we use 10% existing links as test positive links in the test graph. For PPI dataset, we randomly select 3 graphs as training, validation and testing datasets and we sample 10% existing links in the validation/testing graphs as validation/test positive links.

**Baselines & Implementation details.** We use the same baselines as Task 1, while we do not use one-hot features for VGAE, since the training and test graphs have different sizes. As for node features, we randomly project them into the same dimension and then perform row normalization on them. Other settings are the same as Task 1. PE is applied to training and testing graphs separately while PE over testing graphs is computed after removing the testing links.

**Results** are shown in Table 3. Compared with Table 1, for VGAE, we notice that node features perform much worse (except PPI) than Task 1, which demonstrates the risks of using node features when the domain shifts. Positional features, which is not specified for the same graph, is possibly more generalizable over different graphs. Random features are generalizable while still hard to converge. PGNN and GNN Trans. do not utilize positional features appropriately and perform far from ideal. Both SEAL and PEG outperform other baselines significantly, which implies their good stability and generalization. PEG and SEAL again achieve comparable performance while PEG has much lower training and testing complexity. Our results successfully demonstrate the significance of using permutation equivariant and stable PE.

## 5 CONCLUSION

In this work, we studied how GNNs should work with PE in principle, and proposed the conditions that keep GNNs permutation equivariant and stable when PE is used to avoid the node ambiguity issue. We follow those conditions and propose the PEG layer. Extensive experiments on link prediction demonstrate the effectiveness of PEG. In the future, we plan to generalize the theory to more general PE techniques and test PEG over other graph learning tasks.

ACKNOWLEDGMENTS

We greatly thank the actionable suggestions given by reviewers. H. Yin and P. L. are supported by the 2021 JPMorgan Faculty Award and the National Science Foundation (NSF) award HDR-2117997.

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

# A    PROOF OF LEMMA 3.4

Recall the eigenvalues of the PSD matrix $B$ are $0 = \lambda_1 < \lambda_2 < ... < \lambda_p < \lambda_{p+1} \leq \lambda_{p+2} \leq ... \leq \lambda_N$. Suppose one EVD of $B = U\Lambda U^T$. In $U = [u_1, u_2, ..., u_N]$, $u_i$ is the eigenvector of $\lambda_i$ so $Bu_i = \lambda_i u_i$. Without loss of generality, we set $\text{PE}(B) = [u_1, u_2, ..., u_p]$.

Suppose $k = \arg\min_{1 \leq i \leq p} |\lambda_{i+1} - \lambda_i|$. Now, we perturb $B$ by slightly perturbing $u_k$ and $u_{k+1}$. We set

$$u'_k = \sqrt{1 - \epsilon^2} u_k + \epsilon u_{k+1}$$
$$u'_{k+1} = -\epsilon u_k + \sqrt{1 - \epsilon^2} u_{k+1}$$

Set $u'_i = u_i$ for $i \in [N], i \neq k, k+1$. Note that $\|u'_k\| = \|u'_{k+1}\| = 1$ and $u'^T_k u'_{k+1} = 0$. Then, the columns of $U' = [u'_1, u'_2, ..., u'_N]$ still give a group of orthogonal bases.

Now we denote the above perturbation of $B$ as $B + \triangle B = \sum_{i=1}^N \lambda_i u'_i u'^T_i$. Then, $\text{PE}(B + \triangle B)$ could be $[u'_1, u'_2, ..., u'_p]S'$ for any $S' \in \text{SN}(p)$. Therefore, for sufficiently small $\epsilon > 0$,

$$\min_{S' \in \text{SN}(p)} \|\text{PE}(B + \triangle B)S' - \text{PE}(B)\|_F^2$$
$$= \|[(\sqrt{1-\epsilon^2} - 1)u_k + \epsilon u_{k+1}, (\sqrt{1-\epsilon^2} - 1)u_{k+1} - \epsilon u_k]\|_F^2$$
$$= \|(\sqrt{1-\epsilon^2} - 1)u_k + \epsilon u_{k+1}\|^2 + \|(\sqrt{1-\epsilon^2} - 1)u_{k+1} - \epsilon u_k\|^2$$
$$= 4(1 - \sqrt{1-\epsilon^2})$$
$$= 2\epsilon^2 + o(\epsilon^2). \tag{8}$$

Next, we characterize $\|\triangle B\|_F$.

$$\|\triangle B\|_F^2 = \|B + \triangle B - B\|_F^2 = \|\sum_{i=1}^N \lambda_i u'_i u'^T_i - \sum_{i=1}^N \lambda_i u_i u_i^T\|_F^2$$
$$= \|\lambda_k(u'_k u'^T_k - u_k u_k^T) + \lambda_{k+1}(u'_{k+1} u'^T_{k+1} - u_{k+1} u_{k+1}^T)\|_F^2$$
$$= \|(\lambda_{k+1} - \lambda_k)\left[-\epsilon^2(u_k u_k^T - u_{k+1} u_{k+1}^T) + \epsilon\sqrt{1-\epsilon^2}(u_k u_{k+1}^T + u_{k+1} u_k^T)\right]\|_F^2$$
$$= (\lambda_{k+1} - \lambda_k)^2(\epsilon^2 \|u_k u_{k+1}^T + u_{k+1} u_k^T\|_F^2 + o(\epsilon^2))$$
$$= 2(\lambda_{k+1} - \lambda_k)^2(\epsilon^2 + o(\epsilon^2)). \tag{9}$$

Combining Eqs. 8,9, we have, for sufficiently small $\epsilon > 0$,

$$\min_{S' \in \text{SN}(p)} \|\text{PE}(B + \triangle B)S' - \text{PE}(B)\|_F > 0.99|\lambda_{k+1} - \lambda_k|^{-1}\|\triangle B\|_F + o(\epsilon),$$

which concludes the proof.

# B    PROOF OF LEMMA 3.5

The result of the Lemma 3.5 can be derived from the Davis-Kahan theorem (Davis & Kahan, 1970) and its variant (Yu et al., 2015) that characterizes the eigenspace perturbation. We apply the Theorem 2 Eq.3 of (Yu et al., 2015) to two PSD matrices.

**Theorem B.1** (Theorem 2 (Yu et al., 2015)). *Let two PSD matrices* $B^{(1)}, B^{(2)} \in \mathbb{R}^{N \times N}$, *with eigenvalues* $0 = \lambda_1^{(i)} \leq \lambda_2^{(i)} \leq ... \leq \lambda_p^{(i)} \leq ... \leq \lambda_p^{(N)}$ *such that* $\lambda_{p+1}^{(1)} - \lambda_p^{(1)} > 0$. *For*

$i = 1, 2$, let $U^{(i)} = (u_1^{(i)}, u_2^{(i)}, ..., u_p^{(i)})$ have orthonormal columns satisfying $B^{(i)} u_k^{(i)} = \lambda_k u_k^{(i)}$ for $k = 1, 2, ..., p$. Then, there exists an orthogonal matrix $Q \in \mathrm{SO}(p)$ such that

$$\|U^{(2)} Q - U^{(1)}\|_\mathrm{F} \leq \frac{2^{3/2} \min(p^{1/2} \|B^{(1)} - B^{(2)}\|_\mathrm{op}, \|B^{(1)} - B^{(2)}\|_\mathrm{F})}{\lambda_{p+1}^{(1)} - \lambda_p^{(1)}}$$

By symmetry, use the above theorem again and we know there exists $Q' \in \mathrm{SO}(p)$

$$\|U^{(1)} Q' - U^{(2)}\|_\mathrm{F} \leq \frac{2^{3/2} \min(p^{1/2} \|B^{(1)} - B^{(2)}\|_\mathrm{op}, \|B^{(1)} - B^{(2)}\|_\mathrm{F})}{\lambda_{p+1}^{(2)} - \lambda_p^{(2)}}$$

Because $\|U^{(1)} Q' - U^{(2)}\|_\mathrm{F} = \|U^{(2)} Q'^T - U^{(1)}\|_\mathrm{F}$, then there exists $Q \in \mathrm{SO}(p)$,

$$\|U^{(2)} Q - U^{(1)}\|_\mathrm{F} \leq 2^{3/2} \delta \min(p^{1/2} \|B^{(1)} - B^{(2)}\|_\mathrm{op}, \|B^{(1)} - B^{(2)}\|_\mathrm{F}) \tag{10}$$

where $\delta = \min\{(\lambda_{p+1}^{(1)} - \lambda_p^{(1)})^{-1}, (\lambda_{p+1}^{(2)} - \lambda_p^{(2)})^{-1}\}$.

When we apply a permutation matrix $P \in \Pi$ to permute the rows and columns of $B^{(2)}$, then $(PB^{(2)} P^T)(Pu_k^{(2)}) = PB^{(2)} u_k^{(2)} = \lambda_k^{(2)} Pu_k^{(2)}$ for any $k$. Moreover, permuting the rows and columns of a PSD matrix will not change its eigenvalues. This means that $B^{(2)}$ can be replaced by $PB^{(2)} P^T$ in Eq.10 as long as $U^{(2)}$ is replaced by $PU^{(2)}$. Therefore,

$$\|PU^{(2)} Q - U^{(1)}\|_\mathrm{F} \leq 2^{3/2} \delta \min(p^{1/2} \|B^{(1)} - PB^{(2)} P^T\|_\mathrm{op}, \|B^{(1)} - PB^{(2)} P^T\|_\mathrm{F})$$

## C    PROOF OF THEOREM 3.6

To prove PE-equivariance, consider two graphs $\mathcal{G}^{(1)} = (A^{(1)}, X^{(1)})$ and $\mathcal{G}^{(2)} = (A^{(2)}, X^{(2)})$ that have perfect matching $P^*$. So, $L^{(1)} = P^* L^{(2)} P^{*T}$, $X^{(1)} = P^* X^{(2)}$.

Let $Z^{(i)}$ denote the Laplacian eigenmaps of $L^{(i)}$, $i = 1, 2$. Set $B^{(i)} = L^{(i)}$, $i = 1, 2$ and $P = P^*$ and use Lemma 3.5. Because $L^{(1)} = P^* L^{(2)} P^{*T}$ and $\lambda_p \neq \lambda_{p+1}$, then there exists $Q \in \mathrm{SO}(p)$.

$$Z^{(1)} = P^* Z^{(2)} Q \tag{11}$$

Now, we consider a GNN layer $g$ that satisfies Eqs. 4,5. Also denote the output as $(\hat{X}^{(1)}, \hat{Z}^{(1)}) = g(A^{(1)}, X^{(1)}, Z^{(1)})$ and $(\hat{X}^{(2)}, \hat{Z}^{(2)}) = g(A^{(2)}, X^{(2)}, Z^{(2)})$.

$$
\begin{aligned}
(\hat{X}^{(1)}, \hat{Z}^{(1)}) &= g(A^{(1)}, X^{(1)}, Z^{(1)}) \\
&\overset{(a)}{=} g(P^* A^{(2)} P^{*T}, P^{*T} X^{(2)}, P^* Z^{(2)} Q) \\
&\overset{(b)}{=} P^* g(A^{(2)}, X^{(2)}, Z^{(2)} Q) \\
&\overset{(c)}{=} (P^* \hat{X}^{(2)}, P^* \hat{Z}^{(2)} Q)
\end{aligned}
$$

Here (a) is because the perfect matching between $\mathcal{G}^{(1)}$ and $\mathcal{G}^{(2)}$, and Eq. 11. (b) is due to Eq. 4 and (c) is due to Eq. 5.

Therefore, $\hat{X}^{(1)} = P^* \hat{X}^{(2)}$ and $\eta(\hat{Z}^{(1)}, \hat{Z}^{(2)}) = 0$, which implies that $g$ satisfies PE-equivariance.

## D    PROOF OF THEOREM 3.7

To prove PE-stability, consider two graphs $\mathcal{G}^{(1)} = (A^{(1)}, X^{(1)})$ and $\mathcal{G}^{(2)} = (A^{(2)}, X^{(2)})$. Let $P^*$ denote their matching. We study the PEG layer in Eq. 6 and denote $(\hat{X}^{(i)}, \hat{Z}^{(i)}) = g_{\mathrm{PEG}}(A^{(i)}, X^{(i)}, Z^{(i)})$ for $i = 1, 2$.

Let us first bound the easier term regarding the positional features $\eta(\hat{Z}^{(1)}, P^* \hat{Z}^{(2)})$. Because $\hat{Z}^{(i)} = Z^{(i)}$, $\eta(\hat{Z}^{(1)}, P^* \hat{Z}^{(2)}) = \eta(Z^{(1)}, P^* Z^{(2)})$, while the bound of the later is given by Lemma 3.5. Set $B^{(i)} = L^{(i)}$, $i = 1, 2$ and $P = P^*$. Then, there exists a $Q \in \mathrm{SO}(p)$,

$$\|P^* Z^{(2)} Q - Z^{(1)}\|_{\mathrm{F}} \leq 2^{3/2} \delta \min(p^{1/2} \|L^{(1)} - P^* L^{(2)} P^{*T}\|_{\mathrm{op}}, \|L^{(1)} - P^* L^{(2)} P^{*T}\|_{\mathrm{F}}). \quad (12)$$

Next, we bound the harder part $\|\hat{X}^{(1)} - P^* \hat{X}^{(2)}\|_{\mathrm{F}}$. First,

$$\|\hat{X}^{(1)} - P^* \hat{X}^{(2)}\|_{\mathrm{F}} = \|\psi\left[\left(\hat{A}^{(1)} \odot \Xi^{(1)}\right) X^{(1)} W\right] - P^* \psi\left[\left(\hat{A}^{(2)} \odot \Xi^{(2)}\right) X^{(2)} W\right]\|_{\mathrm{F}}. \quad (13)$$

Here, without loss of generality, we set $\Xi_{uv}^{(i)} = \phi(\|Z_u^{(i)} - Z_v^{(i)}\|)$ for $u, v \in [N]$ such that $\hat{A}_{uv}^{(1)} \neq 0$ and otherwise 0. Moreover, $\Xi_{uv}^{(i)}$ is bounded because $\Xi_{uv}^{(i)} = \phi(\|Z_u^{(i)} - Z_v^{(i)}\|) \leq \ell_\phi \|Z_u^{(i)} - Z_v^{(i)}\| \leq \ell_\phi(\|Z_u^{(i)}\| + \|Z_v^{(i)}\|) \leq 2\ell_\phi$ because of the $\ell_\phi$-Lipschitz continuity of $\phi$ and $\|Z_u^{(i)}\| \leq 1$.
Compute the difference

$$\|\psi\left[\left(\hat{A}^{(1)} \odot \Xi^{(1)}\right) X^{(1)} W\right] - P^* \psi\left[\left(\hat{A}^{(2)} \odot \Xi^{(2)}\right) X^{(2)} W\right]\|_{\mathrm{F}}$$

$$\overset{(a)}{\leq} \ell_\psi \|\left(\hat{A}^{(1)} \odot \Xi^{(1)}\right) X^{(1)} W - P^* \left(\hat{A}^{(2)} \odot \Xi^{(2)}\right) X^{(2)} W\|_{\mathrm{F}}$$

$$\overset{(b)}{\leq} \ell_\psi \|W\|_{\mathrm{op}} \|\left(\hat{A}^{(1)} \odot \Xi^{(1)}\right) X^{(1)} - P^* \left(\hat{A}^{(2)} \odot \Xi^{(2)}\right) X^{(2)}\|_{\mathrm{F}}$$

$$\overset{(c)}{=} \ell_\psi \|W\|_{\mathrm{op}} \|\left(\hat{A}^{(1)} \odot \Xi^{(1)} - P^* \left(\hat{A}^{(2)} \odot \Xi^{(2)}\right) P^{*T}\right) X^{(1)} + P^* \left(\hat{A}^{(2)} \odot \Xi^{(2)}\right) (P^{*T} X^{(1)} - X^{(2)})\|_{\mathrm{F}}$$

$$\overset{(d)}{\leq} \ell_\psi \|W\|_{\mathrm{op}} \left[\|\hat{A}^{(1)} \odot \Xi^{(1)} - P^* \left(\hat{A}^{(2)} \odot \Xi^{(2)}\right) P^{*T}\|_{\mathrm{F}} \|X^{(1)}\|_{\mathrm{op}} + \right. \quad (14)$$

$$\left. \|P^{*T} \left(\hat{A}^{(2)} \odot \Xi^{(2)}\right)\|_{\mathrm{op}} \|P^{*T} X^{(1)} - X^{(2)}\|_{\mathrm{F}}\right]$$

$$\overset{(e)}{\leq} \ell_\psi \|W\|_{\mathrm{op}} \left[\|\hat{A}^{(1)} \odot \Xi^{(1)} - P^* \left(\hat{A}^{(2)} \odot \Xi^{(2)}\right) P^{*T}\|_{\mathrm{F}} \|X^{(1)}\|_{\mathrm{op}} + \|\hat{A}^{(2)} \odot \Xi^{(2)}\|_{\mathrm{op}} \|X^{(1)} - P^* X^{(2)}\|_{\mathrm{F}}\right] \quad (15)$$

Here, (a) is due to the $\ell_\psi$-Lipschitz continuity of the component-wise function $\psi$. (b) and (d) use that for any two matrices $A, B$, $\|AB\|_{\mathrm{F}} \leq \min\{\|A\|_{\mathrm{F}} \|B\|_{\mathrm{op}}, \|A\|_{\mathrm{op}} \|B\|_{\mathrm{F}}\}$ and (d) also uses the triangle inequality of $\|\cdot\|_{\mathrm{F}}$. (e) uses $\|PA\|_{\mathrm{op}} = \|A\|_{\mathrm{op}}$ and $\|PA\|_{\mathrm{F}} = \|A\|_{\mathrm{F}}$ for any $P \in \Pi$.

Next, we only need to bound $\|\hat{A}^{(1)} \odot \Xi^{(1)} - P^* \left(\hat{A}^{(2)} \odot \Xi^{(2)}\right) P^{*T}\|_{\mathrm{F}}$ and $\|\hat{A}^{(2)} \odot \Xi^{(2)}\|_{\mathrm{op}}$.

$$\|\hat{A}^{(1)} \odot \Xi^{(1)} - P^* \left(\hat{A}^{(2)} \odot \Xi^{(2)}\right) P^{*T}\|_{\mathrm{F}}$$

$$\overset{(a)}{=} \|\hat{A}^{(1)} \odot \Xi^{(1)} - \left(P^* \hat{A}^{(2)} P^{*T}\right) \odot \left(P^* \Xi^{(2)} P^{*T}\right)\|_{\mathrm{F}}$$

$$\leq \|\hat{A}^{(1)} \odot \left(\Xi^{(1)} - P^* \Xi^{(2)} P^{*T}\right)\|_{\mathrm{F}} + \|\left(\hat{A}^{(1)} - P^* \hat{A}^{(2)} P^{*T}\right) \odot \left(P^* \Xi^{(2)} P^{*T}\right)\|_{\mathrm{F}}$$

$$\overset{(b)}{\leq} \max_{u,v \in [N]} |\hat{A}_{uv}^{(1)}| \|\Xi^{(1)} - P^* \Xi^{(2)} P^{*T}\|_{\mathrm{F}} + \max_{u,v \in [N]} |\Xi_{uv}^{(2)}| \|\hat{A}^{(1)} - P^* \hat{A}^{(2)} P^{*T}\|_{\mathrm{F}}$$

$$\overset{(c)}{\leq} \|\Xi^{(1)} - P^* \Xi^{(2)} P^{*T}\|_{\mathrm{F}} + \ell_\phi \|\hat{L}^{(1)} - P^* \hat{L}^{(2)} P^{*T}\|_{\mathrm{F}} \quad (16)$$

where (a) is because $P(A \odot B) P^T = (P A P^T) \odot (P B P^T)$ for any $P \in \Pi$. (b) is because for any two matrices $A, B$, $\|A \odot B\|_{\mathrm{F}} \leq \max_{u,v} |A_{uv}| \|B\|_{\mathrm{F}}$. (c) is because $\max_{u,v \in [N]} |\hat{A}_{uv}^{(1)}| \leq 1$, $\max_{u,v \in [N]} |\Xi_{uv}^{(2)}| \leq \ell_\phi$.

$$\|\hat{A}^{(2)} \odot \Xi^{(2)}\|_{\mathrm{op}} \overset{(a)}{\leq} \|\hat{A}^{(2)}\|_{\mathrm{op}} \|\Xi^{(2)}\|_{\mathrm{op}} \overset{(b)}{=} \rho(\hat{A}^{(2)}) \rho(\Xi^{(2)}) \overset{(c)}{\leq} 2 \cdot d_{\max}^{(2)} \ell_\phi = 2 d_{\max}^{(2)} \ell_\phi \quad (17)$$

where (a) is because for any two matrices $A, B$, $\|A \odot B\|_{\mathrm{op}} \leq \|A\|_{\mathrm{op}} \|B\|_{\mathrm{op}}$. (b) is because graphs are undirected and both $\hat{A}^{(2)}$ and $\Xi^{(2)}$ are symmetric matrices. Hence, $\hat{A}^{(2)}$ and $\Xi^{(2)}$ can be diagonalized. For diagonalizable matrices, their operator norms equal their spectral radius $\rho$. (c) is because the

following facts: It is known that a degree normalized adjacency matrix $\hat{A}^{(2)}$ has eigenvalues between -1 and 1; And, $\rho(\Xi^{(2)}) \leq \max_{v \in [N]} \sum_{u=1}^{N} |\Xi_{vu}^{(2)}| \leq 2 \max_{v \in [N]} d_v^{(2)} \ell_\phi = 2 d_{\max}^{(2)} \ell_\phi$. Here, we use that $\Xi_{vu}^{(2)}$ is not zero iff $vu$ is an edge in the graph.

Lastly, we need to bound $\|\Xi^{(1)} - P^* \Xi^{(2)} P^{*T}\|_F$ in Eq.16. Let $\pi : [N] \to [N]$ denote the permutation mapping defined by $P^*$, i.e., $P_{uv}^* = 1$ when $v = \pi(u)$ and $P_{uv}^* = 0$ otherwise. Pick the $Q \in \mathrm{SO}(p)$ that matches the two groups of positional features $Z^{(1)}, Z^{(2)}$.

$$\|\Xi^{(1)} - P^* \Xi^{(2)} P^{*T}\|_F$$

$$\overset{(a)}{\leq} \sqrt{\sum_{u,v} \ell_\phi^2 \left( \|Z_u^{(1)} - Z_v^{(1)}\| - \|Z_{\pi(v)}^{(2)} - Z_{\pi(v)}^{(2)}\| \right)^2}$$

$$= \sqrt{\sum_{u,v} \ell_\phi^2 \left( \|Z_u^{(1)} - Z_v^{(1)}\| - \|Z_u^{(1)} - Z_{\pi(v)}^{(2)} Q\| + \|Z_u^{(1)} - Z_{\pi(v)}^{(2)} Q\| - \|Z_{\pi(u)}^{(2)} Q - Z_{\pi(v)}^{(2)} Q\| \right)^2}$$

$$\leq \ell_\phi \sqrt{2 \sum_{u,v} \left( \|Z_u^{(1)} - Z_v^{(1)}\| - \|Z_u^{(1)} - Z_{\pi(v)}^{(2)} Q\| \right)^2 + \left( \|Z_u^{(1)} - Z_{\pi(v)}^{(2)} Q\| - \|Z_{\pi(u)}^{(2)} Q - Z_{\pi(v)}^{(2)} Q\| \right)^2}$$

$$\overset{(b)}{\leq} \ell_\phi \sqrt{2 \sum_{u,v} \left( \|Z_v^{(1)} - Z_{\pi(v)}^{(2)} Q\|^2 + \|Z_u^{(1)} - Z_{\pi(u)}^{(2)} Q\|^2 \right)}$$

$$= 2 \ell_\phi \|Z^{(1)} - P^* Z^{(2)} Q\|_F$$

$$= 2 \ell_\phi \eta(Z^{(1)}, P^* Z^{(2)}) \tag{18}$$

where (a) is due to the $\ell_\phi$-Lipschitz continuity of the component-wise function $\phi$ and (b) is because of triangle inequalities.

Plugging Eq.18 into Eq.16 and plugging Eqs.16,17 into Eq.15, further using Eqs.13,12, we achieve

$$\|\hat{X}^{(1)} - P^* \hat{X}^{(2)}\|_F$$

$$\leq \ell_\psi \ell_\phi \|W\|_{\mathrm{op}} \left[ (2^{5/2}\delta + 1)\|X^{(1)}\|_{\mathrm{op}} \|L^{(1)} - P^* L^{(2)} P^{*T}\|_F + 2 d_{\max}^{(2)} \ell_\phi \|X^{(1)} - P^* X^{(2)}\|_F \right]$$

$$\leq (7\delta \|X^{(1)}\|_{\mathrm{op}} + 2 d_{\max}^{(2)}) \ell_\psi \ell_\phi \|W\|_{\mathrm{op}} d(\mathcal{G}^{(1)}, \mathcal{G}^{(2)}). \tag{19}$$

Combining Eq.19 with the bound on positional features in Eq.12, we conclude the proof by

$$\|\hat{X}^{(1)} - P^* \hat{X}^{(2)}\|_F + \eta(\hat{Z}^{(1)}, P^* \hat{Z}^{(2)})$$

$$\leq [(7\delta \|X^{(1)}\|_{\mathrm{op}} + 2 d_{\max}^{(2)}) \ell_\psi \ell_\phi \|W\|_{\mathrm{op}} + 3\delta] d(\mathcal{G}^{(1)}, \mathcal{G}^{(2)}). \tag{20}$$

# E    PROOF OF THEOREM 3.8

To prove PE-equivariance, consider two graphs $\mathcal{G}^{(1)} = (A^{(1)}, X^{(1)})$ and $\mathcal{G}^{(2)} = (A^{(2)}, X^{(2)})$ that have perfect matching $P^*$, $L^{(1)} = P^* L^{(2)} P^{*T}$, $X^{(1)} = P^* X^{(2)}$.

Let $Z^{(i)}$ denote the positional features obtained by decomposing the optimal solution $M^{(i)*}$ to the optimization problem Eq.7. Because $L^{(1)} = P^* L^{(2)} P^{*T}$, we have $A^{(1)} = P^* A^{(2)} P^{*T}$ and $D^{(1)} = P^* D^{(2)} P^{*T}$. Then,

$$M^{(1)*} = \underset{M : \mathrm{rank}(M) \leq p}{\arg\min} \; \mathrm{tr}(f_+(A^{(1)})g(M) + f_-(D^{(1)})g(-M))$$

$$\overset{(a)}{=} \underset{M : \mathrm{rank}(M) \leq p}{\arg\min} \; \mathrm{tr}(P^* f_+(A^{(2)}) P^{*T} g(M) + P^* f_-(D^{(2)}) P^{*T} g(-M))$$

$$\overset{(b)}{=} \underset{M : \mathrm{rank}(M) \leq p}{\arg\min} \; \mathrm{tr}(f_+(A^{(2)}) P^{*T} g(M) P^* + f_-(D^{(2)}) P^{*T} g(-M) P^*)$$

$$\overset{(c)}{=} \underset{M : \mathrm{rank}(M) \leq p}{\arg\min} \; \mathrm{tr}(f_+(A^{(2)}) g(P^{*T} M P^*) + f_-(D^{(2)}) g(-P^{*T} M P^*))$$

$$\overset{(d)}{=} P^* M^{(2)*} P^{*T}$$

Here (a) is because $A^{(1)} = P^* A^{(2)} P^{*T}$ and the assumptions on $f_+$ and $f_-$. (b) is because for two squared matrices $A, B$, $\text{tr}(AB) = \text{tr}(BA)$. (c) is because $g$ is component-wise function. (d) is because $M^{*(2)}$ is the unique solution of $\arg\min_{M:\text{rank}(M)\leq p} \text{tr}(f_+(A^{(2)})g(M) + f_-(D^{(2)})g(-M))$.

Recall $M^{(1)*} = Z'^{(1)} Z^{(1)T}$, $Z^{(1)T} Z^{(1)} = I$ and $M^{(2)*} = Z'^{(2)} Z^{(2)T}$, $Z^{(2)T} Z^{(2)} = I$. Note that $Z^{(1)}, Z^{(2)}$ that satisfy such decompositions are not unique. As $\text{rank}(M^{(1)*}) = \text{rank}(M^{(2)*}) = p$, so $Z^{(1)}, Z'^{(1)}$ and $P^* Z^{(2)}, P^* Z'^{(2)}$ have full-rank columns.

Since $M^{(1)*} = P^* M^{(2)*} P^{*T}$, $Z'^{(1)} Z^{(1)T} = P^* Z'^{(2)} Z^{(2)T} P^{*T}$. Because $Z'^{(1)}$ has full-rank columns, $Z'^{(1)T} Z'^{(1)}$ is non-singular. Let $Q = Z'^{(2)T} P^{*T} Z'^{(1)} (Z'^{T(1)} Z'^{(1)})^{-1}$. Then, $Z^{(1)} = P^* Z^{(2)} Q$. Since, $Z^{(1)T} Z^{(1)} = Z^{(2)T} Z^{(2)} = I$, we have $Q^T Q = I$. $Q$ is a squared matrix so $Q \in \text{SO}(p)$. That means $Z^{(1)} = P^* Z^{(2)} Q$ for some $Q \in \text{SO}(p)$.

Now, we consider a GNN layer $g$ that satisfies Eqs. 4,5. Also denote the output as $(\hat{X}^{(1)}, \hat{Z}^{(1)}) = g(A^{(1)}, X^{(1)}, Z^{(1)})$ and $(\hat{X}^{(2)}, \hat{Z}^{(2)}) = g(A^{(2)}, X^{(2)}, Z^{(2)})$.

$$
\begin{aligned}
(\hat{X}^{(1)}, \hat{Z}^{(1)}) &= g(A^{(1)}, X^{(1)}, Z^{(1)}) \\
&\overset{(a)}{=} g(P^* A^{(2)} P^{*T}, P^{*T} X^{(2)}, P^* Z^{(2)} Q) \\
&\overset{(b)}{=} P^* g(A^{(2)}, X^{(2)}, Z^{(2)} Q) \\
&\overset{(c)}{=} (P^* \hat{X}^{(2)}, P^* \hat{Z}^{(2)} Q)
\end{aligned}
$$

Here (a) is because the perfect matching between $\mathcal{G}^{(1)}$ and $\mathcal{G}^{(2)}$, and $Z^{(1)} = P^* Z^{(2)} Q$. (b) is due to Eq. 4 and (c) is due to Eq. 5.

Therefore, $\hat{X}^{(1)} = P^* \hat{X}^{(2)}$ and $\eta(\hat{Z}^{(1)}, \hat{Z}^{(2)}) = 0$, which implies that $g$ satisfies PE-equivariance.

# F    REVIEW OF THE E-GNN LAYER

Satorras et al. (2021) studies the problem when the nodes of a graph have physical coordinates as features and proposes E-GNN to deal with this kind of graph data. E-GNN aims to keep permutation equivariant with respect to the node order, and translation equivariant, rotation equivariant with respect to the physical coordinate features. As E-GNN asks even more (translation equivariance) than Eqs. 4,5, E-GNN can be adopted to leverage PE techniques to keep PE-equivariance. The specific form of E-GNN is as follows.

Given a graph $\mathcal{G} = (\mathcal{V}, \mathcal{E})$ with nodes $v_i \in \mathcal{V}$ and edges $e_{ij} \in \mathcal{E}$. E-GNN layer takes the node embeddings $\mathbf{h}^l = \{\mathbf{h}_0^l, ..., \mathbf{h}_{M-1}^l\}$, coordinate embeddings $\mathbf{x}^l = \{\mathbf{x}_0^l, ..., \mathbf{x}_{M-1}^l\}$ and edge information $\mathcal{E} = (e_{ij})$ as input and outputs $\mathbf{h}^{l+1}$ and $\mathbf{x}^{l+1}$, respectively. Thus, the E-GNN layer can be denoted as: $\mathbf{h}^{l+1}, \mathbf{x}^{l+1} = \text{EGCL}(\mathbf{h}^l, \mathbf{x}^l, \mathcal{E})$. The layer is defined as following:

$$
\begin{aligned}
\mathbf{m}_{ij} &= \phi_e(\mathbf{h}_i^l, \mathbf{h}_j^l, ||\mathbf{x}_i^l - \mathbf{x}_j^l||^2, a_{ij}) \\
\mathbf{x}_i^{(l+1)} &= \mathbf{x}_i^l + \sum_{j \neq i} (\mathbf{x}_i^l - \mathbf{x}_j^l) \phi_x(\mathbf{m}_{ij}) \\
\mathbf{m}_i &= \sum_{j \in \mathcal{N}_i} (\mathbf{m}_{ij}) \\
\mathbf{h}_i^{(l+1)} &= \phi_h(\mathbf{h}_i^{(l)}, \mathbf{m}_i)
\end{aligned}
$$

Where $a_{ij}$ is the edge attributes, $\phi_e$ represents edge operation, $\phi_x$ represents edge embedding operation and $\phi_h$ represents node operation.

# G    THE OPTIMIZATION FORM FOR GENERAL PE TECHNIQUES

Given an undirected and weighted network $\mathcal{G} = (V, E, A)$ with $N$ nodes, LINE (Tang et al., 2015) with the second order proximity (aka LINE (2nd)) aims to extract two latent representation matrices $Z, Z' \in \mathbb{R}^{N \times p}$. Let $Z_i, Z'_i$ denote the $i$th row of $Z$ and the $i$th row of $Z'$, respectively. The objective function of LINE (2nd) follows

$$\max_{Z,Z'} \quad \sum_{i=1}^{N}\sum_{j=1}^{N} A_{ij}g(Z_i Z'^T_j) + b\sum_{i=1}^{N} \mathbb{E}_{j'\sim\mathbb{P}_V}(g(-Z_i Z'^T_j))$$

where $g(x)$ is the log sigmoid function $g(x) = x - \log(1 + \exp(x))$ and $b$ is a positive constant. The first term corresponds to the positive examples, i.e., links in the graph while the second term is based on network negative sampling. Also, $\mathbb{P}_V$ is some distribution defined over the node set. LINE adopts $\mathbb{P}_V(j) \propto d_j^{\frac{3}{4}}$ where $d_j$ is the degree of node $j$. By filling the expectation and using matrix form to rewrite the objective, we have

$$\sum_{i=1}^{N}\sum_{j=1}^{N} A_{ij}g(Z_i Z'^T_j) + b\sum_{i=1}^{N} \mathbb{E}_{j'\sim\mathbb{P}_V}(g(-Z_i Z'^T_j))$$
$$=\sum_{i=1}^{N}\sum_{j=1}^{N} A_{ij}g(Z_i Z'^T_j) + c\sum_{i=1}^{N} d^{\frac{3}{4}}g(-Z_i Z'^T_{j'})$$
$$=\text{tr}(A^T g(ZZ')) + \text{tr}(cD^{\frac{3}{4}}\mathbf{1}\mathbf{1}^T g(ZZ'))$$
$$=\text{tr}(Ag(Z'Z)) + c\mathbf{1}\mathbf{1}^T D^{\frac{3}{4}}g(Z'Z))$$

where $c = \frac{b}{\sum_{j=1}^{N} d_j^{3/4}}$. So for LINE, $f_+(A) = A$ and $f_-(D) = c\mathbf{1}\mathbf{1}^T D^{\frac{3}{4}}$. It is easy to validate that for all $P \in \Pi$, $f_+(PAP^T) = Pf_+(A)P^T$ and $f_-(PDP^T) = Pf_-(D)P^T$, which satisfies the condition in Theorem 3.8.

Deepwalk (Perozzi et al., 2014) can be rewritten by following the similar idea. Deepwalk firstly performs random walks with certain length for many times starting from each node and then treat each walk as a sequence of node-id strings. Deepwalk trains a skip-gram model on these node-id strings (Mikolov et al., 2013). Now we consider the skip-gram model with negative sampling (SGNS). Denote the collection of observed words and their context pairs as $\mathcal{D}$. $\#(w, c)$ denotes the number of times that the pair $(w, c)$ appears in $\mathcal{D}$. $\#(w)$ and $\#(c)$ indicates the number of times $w$ and $c$ occurred in $\mathcal{D}$. Let $Z_w$ denote the vector representation of $w$ and $Z'_c$ denote the vector representation of $c$. According to (Levy & Goldberg, 2014), the objective function of SGNS follows

$$\sum_{w}\sum_{c} \#(w,c)g(Z_w Z'_c) + b\sum_{w} \#(w)\mathbb{E}_{c'\sim\mathbb{P}_c}(g(-Z_w Z'^T_{c'}))$$

Deepwalk adopts this objective by viewing each node $v$ in $V$ as a word and viewing any node that gets sampled simultaneously with $v$ within $T$ hops of the random walk as the context $c$. Let $\Phi = D^{-1}A$ denote the random walk matrix of the graph. In this case, for a node $v$ as the word and for another node $u$ as the context, the expected number $\#(v, u) \propto \Phi'_{vu} = \sum_{k=1}^{T}(d_v(\Phi^k)_{vu} + d_u(\Phi^k)_{uv})$. The expected number $\#(v) \propto d_v$ which uses the stationary distribution of random walk over a connected graph is proportional to the node degrees. We also set the negative context sampling probability $\mathbb{P}_c$ is proportional to the node degrees, i,e., $\mathbb{P}_c(u) \propto d_u$. Then, in Deepwalk, the SGNS objective reduces to

$$\sum_{v=1}^{N}\sum_{u=1}^{N} \Phi'_{vu}g(Z_v Z'_u) + c\sum_{v=1}^{N}\sum_{u=1}^{N} d_v d_u g(-Z_v Z'^T_u)$$

for some positive constant $c$. Similar to the derivation for LINE, we can rewrite it into the matrix form

$$\text{tr}(\Phi' g(Z'Z^T) + D\mathbf{1}\mathbf{1}^T D g(-Z'Z^T)).$$

Table 4: Summary of Twitch dataset

|          | DE      | EN     | ES     | FR      | PT     | RU     |
|----------|---------|--------|--------|---------|--------|--------|
| Nodes    | 9,498   | 7,126  | 4,648  | 6,549   | 1,912  | 4,385  |
| Edges    | 153,138 | 35,324 | 59,382 | 112,666 | 31,299 | 37,304 |
| Features | 3,170   | 3,170  | 3,170  | 3,170   | 3,170  | 3,170  |

Therefore, for Deepwalk, $f_+(A) = \Phi' = \sum_{k=1}^{T}(D\Phi^k + \Phi^{Tk}D)$ and $f_-(D) = cD\mathbf{1}\mathbf{1}^T D$, where $\Phi = D^{-1}A$ and c is a positive constant. It is also easy to verify that for all $P \in \Pi$, $f_+(PAP^T) = Pf_+(A)P^T$ and $f_-(PDP^T) = Pf_-(D)P^T$, which satisfies the conditions in Theorem 3.8.

As for Node2vec (Grover & Leskovec, 2016), it performs a 2nd-order random walk to collect node-id strings and then train an SGNS model. The optimization form is more involved, interested readers could check (Qiu et al., 2018; Grover & Leskovec, 2016) for more details.

# H    SUPPLEMENT FOR EXPERIMENTS

We put more specifics of datasets and baselines adopted in Sec. H.1 and Sec. H.2, respectively. We describe how we tune our model in Sec. H.3. We list further experimental results in Sec. H.4. We further conduct three supplementary experiments t demonstrate the generalization capability, stability and wide applicability of our theory and proposed PEG layer in Secs. H.6-H.8.

## H.1    DATASETS

The citation networks-**Cora**, **Citeseer** and **Pubmed** are collected by Sen et al. (2008), where nodes represent documents and edges (undirected) represent citations. Node features are the bag-of-words representation of documents. The Cora dataset contains 2708 nodes, 5429 edges and 1433 features per node. The Citeseer dataset contains 3327 nodes, 4732 edges and 3703 features per node. The Pubmed dataset contains 19717 nodes, 44338 edges and 500 features per node.

**Twitch** is obtained from Rozemberczki et al. (2021). Twitch is a user-user networks of gamers, where nodes correspond to users and edges correspond to mutual friendship between them. Node features correspond to the games they played and liked, users' location and streaming habits. Twitch contains 7 user networks over different countries, including Germany (DE), England (EN), Spain (ES), France (FR), Portugal (PT) and Russia (RU). The details of these datasets are shown in Table 4.

**Chameleon** is used in Chien et al. (2021). Chameleon is a page-page network on topic 'Chameleon' in Wikipedia (December 2018), where nodes correspond to articles and edges represent mutual links between the articles. Node features indicate the presence of several informative nouns in the articles and the average monthly traffic (October 2017 - November 2018). Chameleon dataset contains 2277 nodes, 36101 edges and 2325 features per node.

Protein-protein interaction (PPI) dataset contains 24 graphs corresponding to different human tissues (Zitnik & Leskovec, 2017). We adopt the preprocessed data provided by Hamilton et al. (2017) to construct graphs. The average number of nodes per graph is 2372 and each node has 50 features, which correspond to positional gene sets, motif gene sets and immunological signatures.

Ogbl-ddi and ogbl-collab are chose from the open graph benchmark (OGB) (Hu et al., 2021), which adopt more realistic train/validation/test splitting, such as by time (ogbl-collab) and by by drug target in the body (ogbl-ddi). ogbl-ddi is a drug-drug interaction network, where nodes represent drugs and edges represent interaction between drugs, where the joint effect of taking the two drugs together is considerably different from the effect that taking either of them independently. Ogbl-collab is an author collaboration graph, where nodes represent authors and edges represent the collaboration between authors. The 128 dimension node features of ogbl-collab is extracted by by averaging the word embeddings of the authors' papers. Ogbl-ddi has 4267 nodes and 1.3M edges. Obdl-collab has 0.23M nodes and 1.3M edges.

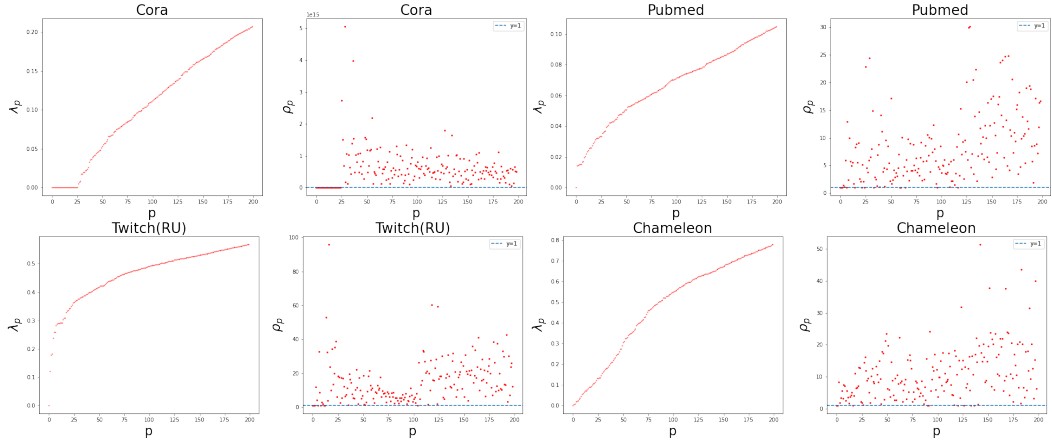

Figure 3: Eigenvalues $\lambda_p$ and the stability ratio $\rho_p = \frac{|\lambda_p - \lambda_{p+1}|}{\min_{1 \le k \le p} |\lambda_k - \lambda_{k+1}|}$. PEG with sensitivity $|\lambda_p - \lambda_{p+1}|^{-1}$ is far more stable than previous methods with sensitivity $\max_{1 \le k \le p} |\lambda_k - \lambda_{k+1}|^{-1}$. Over Cora, $\rho_p$ is extremely large because there are multiple eigenvalues ($\rho_p$ is still finite due to the numerical approximation of the eigenvalues). Over Pubmed, Twitch(RU) and Chameleon even if there are no multiple eigenvalues, $\rho_p$ is mostly larger than 5.

## H.2    BASELINE DETAILS

We have 4 baselines based on GNNs and 2 baselines based on network embedding techniques, namely, LE (Belkin & Niyogi, 2003) and Deepwalk (DW) (Perozzi et al., 2014). We will first introduce the implementation of LE and DW, then discuss other baselines.

Both LE and DW embed the networks in $\mathbb{R}^{128}$ in an unsupervised way. For LE, we factorize of the graph Laplacian matrix: $\Delta = I - D^{-\frac{1}{2}} A D^{-\frac{1}{2}} = U \Lambda U^T$, where $A$ is the adjacency matrix, $D$ is the degree matrix, and $\Lambda$, $U$ correspond respectively to the eigenvalues and eigenvectors. We use the 128 smallest eigenvectors as LE. For DW, we use the code provide by OpenNE[2]. Both methods use the inner product between pairwise node representations as the link representations. Then, the link representations are fed to an MLP for final predictions.

Regarding to GNN-based baselines, VGAE is implemented according to the code[3] (Kipf & Welling, 2016) given by the original paper, with 2 message passing layers with 32 hidden dimensions. P-GNN is implemented by adopting the code[4] provided by the original paper (You et al., 2019), with 2 message passing layers with 32 dimensions, with a tuned dropout ratio in {0, 0.5}. GNN transformer layer is implemented by adopting the code[5] provided by the original paper (Dwivedi & Bresson, 2020) with 2 message passing layers with 128 hidden dimensions. SEAL is implemented by adapting the code[6] provided by the original paper. Notice the ogbl-ddi graph contains no node features, so the we use free-parameter node embeddings as the input node features and train them together with the GNN parameters. We slightly tune the hidden dimensions and layers of these baselines and present the best results.

Moreover, we aim to understand whether PEG is sensitive to the dimension of positional features. We conduct experiments on Cora, Pubmed and Twitch(RU) to test the sensitivity of the dimension positional features. In Fig. 4, we compare the performance of PEG-DW, PEG-LE, PEG-DW+ and PEG-LE+ with PE in different dimensions. All the experiment settings follow the setting to get Table 1 except for the dimension of positional features.

## H.3    HYPERPARAMETERS TUNING FOR PEG

Table 5 lists the most important hyperparameters, which applies to our proposed model PEG. Grid research is used to find the best hyperparameter combination. Note that our model actually got very slightly tuned. We believe more extensively hyperparameter tuning may yield even better results.

---

[2]https://github.com/thunlp/OpenNE/tree/pytorch
[3]https://github.com/tkipf/gae
[4]https://github.com/JiaxuanYou/P-GNN
[5]https://github.com/graphdeeplearning/graphtransformer
[6]https://github.com/muhanzhang/SEAL

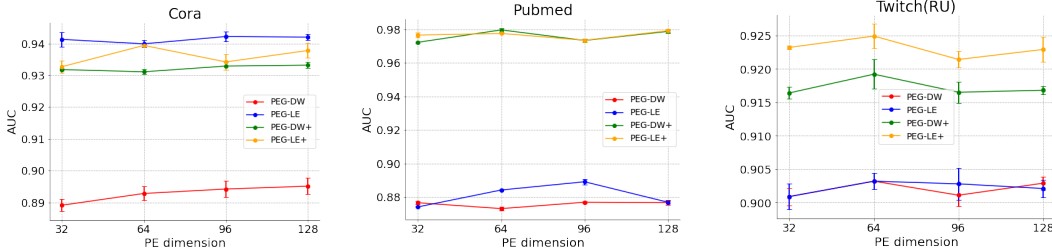

Figure 4: The performance (AUC) of PEG with different dimensions of positional encodings for traditional link prediction (Task 1).

The models are trained until the loss converges and we report the best model by running each for 10 times. We provide our code for reproducing the experimental results to the reviewers and area chairs in the discussion forums.

Table 5: List of hyperparameters and their value/range

| Hyperparameters | Value/Range |
|---|---|
| batch size | 64, 128, 64×1024(ogb) |
| learning rate | 1e-2, 1e-3 |
| optimizer | Adam |
| conv. layers | 2 |
| conv, hidden dim | 128 |
| PE. dim | 128 |

## H.4 FURTHER ANALYSIS

As a supplement to the Lemma 3.4 and Lemma 3.5, we compute the ratio between the inverse eigengap between the $p$th and $(p+1)$th eigenvalues $\{|\lambda_p - \lambda_{p+1}|^{-1}\}$ and $\max_{1 \leq k \leq p} |\lambda_k - \lambda_{k+1}|^{-1}$ over more graphs as shown in Fig. 3.

Fig. 4 shows that different PE techniques have different dimensional sensitivity. In general, DW seems to be more stable than LE over the three datasets and PEG-DW+ achieves stable performance. When the dimension increases, PEG are more likely to achieve higher performance, but will be more time consuming.

## H.5 EDGE WEIGHT VISUALIZATION

In the PEG layer, we calculate an edge weight according to the distance between the representations of the end nodes of the edge. To further understand the learnt relationship between edge weights and the distance between node representations, we visualize the edge weight transformation curves. We run the experiment traditional link prediction (task 1) and use DW as the node positional embedding, i.e., the model PEG-DW. For each dataset, we draw the edge weight transformation curves in Fig. 5. Note that each curve ranges in x-axis from the smallest distance to the largest distance observed from the corresponding graph, we also randomly select 500 distances from each graph and scatter them on the curve as shown in Fig. 5. Fig. 5 shows that the edge weight increases monotonically with respect to the input distance, but the relationship between them is non-linear. Most of the weights are close to 1 but a few weights are much smaller.

## H.6 LINK PREDICTION OVER RANDOM GRAPHS

To further demonstrate the generalization of our model, we conduct inductive link prediction over random graph models. Specifically, we use stochastic block models (SBM) with two blocks Holland et al. (1983) to generate random graphs. Each block contains 500 nodes and the probability to form a link between two nodes within each block is 0.3, while the probability to form a link between two nodes across different blocks is 0.1. Here, we randomly select links inside the blocks as positive

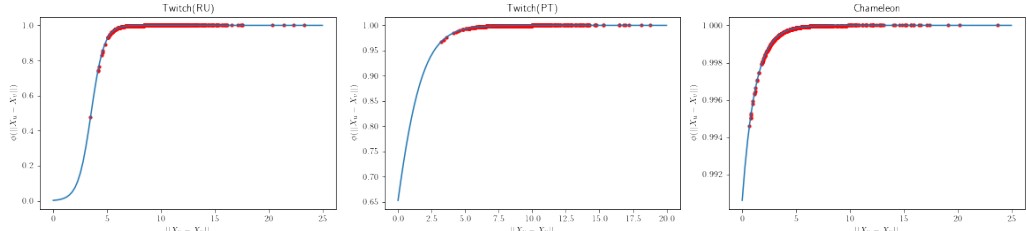

Figure 5: The edge weight transformation curve of PEG for each dataset. 500 selected edges are represented as red points.

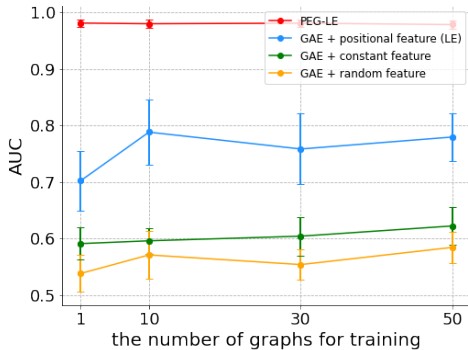

Figure 6: The comparison of AUC-ROC between PEG and GAE baselines for link prediction on random graphs.

samples, and randomly select the missing links (unconnected node pairs) from the generated graph as negative samples. The model is trained by using various numbers of graphs (e.g. 1, 10, 30, 50), validated and tested on 10 random graphs respectively. For each graph that is used for training or testing, we utilize 10% positive links and pair them with the same number of negative missing links for training or testing respectively. The rest settings are remained the same as in Task 2. We choose three variants of GAE that use constant features (node degree), random features and positional features as baselines. All the models are trained until they converge and the models with the best validation performance are used to report the results (averaged by 10 independent runs), which are shown in Fig. 6.

For various sizes of input graphs, PEG achieves AUC around 0.98 by just using one training graph and slightly improves AUC by using more training graphs, which consistently outperforms all three GAE variants that even use the entire 50 training graphs with large margins. This directly demonstrates the stability and better generalization of PEG models.

## H.7 THE STUDY OF PE STABILITY BY PERTURBING THE INPUT GRAPHS

To further investigate PE stability, we consider two versions of models utilizing PE – one is PEG, which is PE equivariant and stable, and the others are GAE/VGAE (N./C. + P.), which fail to satisfy the two equivariant properties and thus may not be stable. These two versions of models are evaluated on the Cora dataset. All the models are trained over the original graph. Then, we perturb the input graph by randomly adding or dropping a certain percentage of links during the inference. The rest of the settings remain the same for link prediction. Specifically, after training the model, we inject a perturbation on top of the input graph by adding extra links (negative samples) or dropping positive links with the ratio of 10%, 20% and 30%, respectively. We recompute the positional encodings to reflect the structural perturbation of the given graph and plug in the perturbed positional encodings into different models. Other experiment settings remain the same as in Task 1. The results are summarized and plotted in Fig. 7.

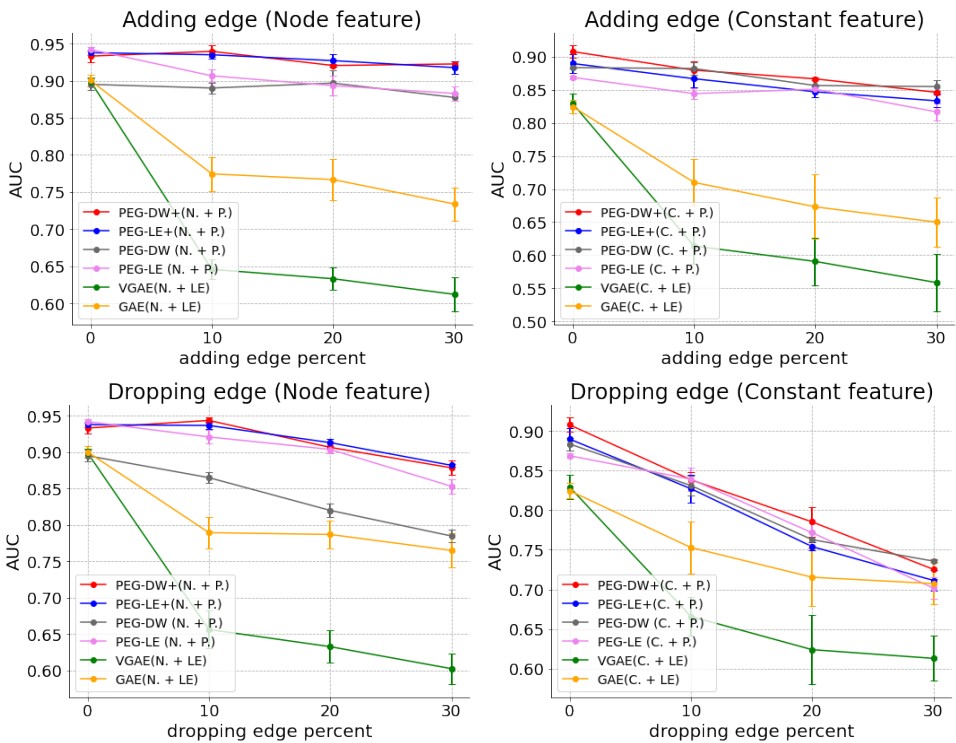

Figure 7: The AUC-ROC of PEG and GAE/VGAE for link prediction in terms of PE stability (w. node or constant feature and w. adding or dropping edges). To perturb the input graphs, we can randomly add or delete some edges with the percentage shown in the X-axis before the inference.

By comparing the models with different levels of input perturbation, we can observe from Fig. 7 that PEG achieves consistently and significantly better performance with either node features or constant features in all four scenarios, which is far more robust than GAE/VGAE-based models that adopt PE under the same level of perturbations. Compared with adding edges, removing edges is more destructive for small-size graphs such as Cora, and thereby does more harm to the model performance. Particularly, all the models using C. + P. achieve subpar performance when considerable number of edges are removed. However, PEG can still show strong resilience against this type of perturbation (especially with node features), while GAE/VGAE generally collapses. These results further support that the performance boost indeed benefits from the equivariance and stability with PEs instead of simply using PE variants.

## H.8 PRELIMINARY RESULTS FOR NODE CLASSIFICATION BASED ON PE

In order to evaluate PEG on a wider category of tasks, we consider node classification on citation networks – **Cora**, **Citeseer** and **Pubmed** (Sen et al., 2008) and closely follow the transductive experimental setup of Yang et al. (2016). Suppose the final readout of PEG is denoted as $(\hat{X}, Z)$. To classify the node $u$, we only utilize its node representation $\hat{X}_u$ to make the final predictions. The rest parts and the hyperparameters of the model are kept the same as the one that is good for our task 1 (traditional link prediction). We did not specifically tune the model for node classification. The results of evaluation on node-level are summarized in Table 6. We report the mean accuracy with standard deviation of classification on the test set of our method after 10 runs.

Both PEG-LE and PEG-DW significantly outperform GCN and provide comparable results against GAT, which illustrates the effectiveness of the proposed PEG on node level tasks. Recall that we almost did not tune our model to achieve such results. Even better results may be expected by further finer tuning the model on each dataset. Moreover, GAT uses multi-head attention to aggregate information from neighbours where the attention weights are based on node features, while PEG only adopts GCN layers with some edge weights derived from positional encodings that merely depend on the network structures. PEG and GAT essentially utilize orthogonal information source to re-weight

Table 6: Summary of results in terms of node classification accuracy (mean ± std%), for Cora, Citeseer and Pubmed.

| Method | Cora | Citeseer | PubMed |
|--------|------|----------|--------|
| PEG-DW | 82.24 ± 0.02 | 71.22 ± 0.01 | 79.83 ± 0.02 |
| PEG-LE | 82.16 ± 0.02 | 71.93 ± 0.01 | 79.85 ± 0.01 |
| GCN | 81.50 ± 0.05 | 70.38 ± 0.05 | 79.03 ± 0.03 |
| GAT | 83.03 ± 0.07 | 72.52 ± 0.07 | 79.06 ± 0.03 |

the edges. Hence, PEG should be easily further combined with the attention mechanism based on node features to get even better predictions, which this is out of the scope of this work, so we leave it for future study.

