# OpenReview forum: "Equivariant and Stable Positional Encoding for More Powerful Graph Neural Networks"
_ICLR.cc/2022/Conference — ICLR 2022 Poster_

### Official Review · Reviewer_KJUw · 2021-10-30

**Correctness:** 4
**Technical Novelty And Significance:** 3
**Empirical Novelty And Significance:** 2
**Recommendation:** 8
**Confidence:** 4

**Main Review:**

Strengths:

1) Lemma 3.4, where the authors theoretically show the particular PE design (of matrix B) violates PE-stability is very useful. Theorem 3.7 where the authors have explicitly derived the value of the constant C for equation 1 is particularly relevant (although the bound can be made tighter).

2) The authors have also put some effort into pointing out the generalizations to other methods. Further, they provide some guarantees for PE methods that follow the optimization objective in equation 7 and show their PE equivariance.

3) Figures 2 and 3 have interesting observations on real world data.

4) I found the Domain-shift link prediction experiments to be very practical and highly interesting.

5) The design choice for the PEG layer in equation 6 is sufficiently simple and easy to compute. The conditions in equation 4 and 5 are very sound. Table 2 shows that the proposed method is also substantially faster than the best performing baseline SEAL.


Weaknesses (and questions for authors):

1) “A GNN layer is expected to be permutation equivariant and stable. These two properties, if satisfied, guarantee that the GNN layer is transferrable and have better generalization performance” - the authors do provide an intuition in subsequent lines and experiments to support their claim in section 4.3, but are there any relevant works that they can direct to? Can, or more concretely, does it hold for any other type of generalization?

2) Can the experiments be done on a wider category of tasks? Although the authors do mention it as a future work, it will be quite interesting to see some evaluation on at least one of: node / graph classification / community detection tasks as well.

3) Can the authors also consider at least one more recent and strong baseline, such as [1]? The evaluation is quite thorough and I understand that it might be difficult to perform these experiments, but it will be quite interesting to see the outcomes, since at least one of the recent methods, ie SEAL, is performing very competitively (and [1] has been shown to outperform SEAL in some cases).

4) Can the authors provide more intuition on why they arrived at the proposed choice in Equation 6? Also, is there any other reason for choosing GCN apart from its simplicity and ease of incorporating the permutation equivariance?

5) The zig-zag patterns in figure 4 are very interesting. I understand that it may or may not be trivial, but it will be helpful if the authors can provide some more intuition for such behavior apart from what is stated in section H.4 .

6) Can the authors elaborate the related works section 1.1 a bit more? Particularly, more details on the methods that propose an explicit use of PE techniques such as [2] and [3] might be helpful.

7) Although intuitively sound, as I have mentioned in the strengths subsection above, it will be great if the authors can provide some reference or more conclusive evidence to this statement - “Inspired by the stability of the eigenspace, the idea to achieve PE-stability is to make the GNN layer invariant to the selection of bases of the eigenspace for the positional features” on the second last paragraph on page 5.

[1] S. Abu-El-Haija, B. Perozzi, R. Al-Rfou, and A. A. Alemi, “Watch your step: Learning node embeddings via graph attention,” in Advances in Neural Information Processing Systems, 2018

[2] Srinivasan, Balasubramaniam, and Bruno Ribeiro. "On the equivalence between positional node embeddings and structural graph representations." arXiv preprint arXiv:1910.00452 (2019)

[3] Dwivedi, Vijay Prakash, and Xavier Bresson. "A generalization of transformer networks to graphs." arXiv preprint arXiv:2012.09699 (2020).

**Summary Of The Paper:**

The authors propose a novel PEG layer to account for positional encodings in graph neural networks and state some necessary conditions for its stability and permutation equivariance while providing theoretical insights for the same. They show that their proposed layer is PE-equivariant under some conditions (theorem 3.6) and also derive the value of the constant C for equation 1 to guarantee PE-stability. Extensive experiments done for the task of link prediction clearly show that the proposed layer is indeed useful and the only subpar baseline method is SEAL (which outperforms PEG in some cases). The authors have also performed experiments to vividly show the generalization ability of their layer using the task of domain-shift link prediction, which are indeed interesting. The simplicity of their approach complemented with a reasonable runtime further adds to the overall utility of this work. The tweak used in section 4.1 for training on graphs where the union of link sets for training, validation and testing cover the entire graph is also useful and can also be leveraged in other methods having explicit positional encodings for GNNs.

Given the recent upsurge in the amount of work done to incorporate positional encodings in GNNs, work such as this that addresses the shortcomings of previous methods and also provides a theoretical framework was necessary.

Small typo on page 4 and 5 - “features” spelled as “feautures” .


**Summary Of The Review:**

Overall the paper is very well written, has some interesting theoretical insights and very practical results. It will be very helpful if the authors can further address some of the  questions and weaknesses as pointed out in the previous subsection of this review. I believe that the authors should at least perform more diverse experiments to completely support their arguments regarding generalization and higher stability. After addressing these minor pointers, I believe that the contributions will be worth presenting at the conference. I additionally support the simplicity of the proposed approach, both analytically and computationally.

---

> ### Author Response · Authors · 2021-11-23
> **Response to Reviewer KJUw (Part I)**
>
> We greatly appreciated Reviewer KJUw for recognizing our theoretical contributions, the interesting aspects of our experiments and strongly supporting the acceptance of this work. Here, we address the questions raised by Reviewer KJUw.
>
> >  "'A GNN layer is expected to be permutation equivariant and stable. These two properties, if satisfied, guarantee that the GNN layer is transferrable and have better generalization performance' - the authors do provide an intuition in subsequent lines and experiments to support their claim in section 4.3, but are there any relevant works that they can direct to? Does, or more concretely, can it hold for any other type of generalization?"
>
> Fundamental NN architectures are proposed to capture equivariance/invariance. CNNs and RNNs are proposed to capture translation equivariance/invariance [1]. GNNs are proposed to capture permutation equivariance/invariance [2]. Actually, very recently, statistical machine learning researchers have proved that the model that captures the equivariance/invariance captured by CNNs can effectively decrease the needed sample complexity [3]. This is for the general notion of generalization instead of a particular type of generalization.
>
> > "Can the experiments be done on a wider category of tasks? Although the authors do mention it as future work, it will be quite interesting to see some evaluation on at least one of node/graph classification / community detection tasks as well."
>
> In the revised manuscript (See Appendix I.1), we have done experiments to predict the links in the blocks of random graphs generated from stochastic block models. This task is also related to community detection. Moreover, we have conducted some preliminary experiments on the node classification task (See Appendix I.2).
>
> > "Can the authors also consider at least one more recent and strong baseline, such as [4]? The evaluation is quite thorough and I understand that it might be difficult to perform these experiments, but it will be quite interesting to see the outcomes, since at least one of the recent methods, ie SEAL, is performing very competitively (and [4] has been shown to outperform SEAL in some cases)."
>
> We have included SEAL as one of the SOTA baselines in our experiments. [4] proposed a family of graph attention models, which can learn graph embeddings through trainable attention parameters instead of manual tuning hyperparameters of context distribution.
> Considering such a limited time for preparing the response and the unavailability of the code in the original paper, we were not able to implement and compare the baseline model [4] in our response.
>
> Instead, in the updated version, we focused on link prediction on random graphs and perturbed graphs, and node classification tasks, to comprehensively illustrate the generalizability, stability, and applicability of our model for various types of graph tasks. Meanwhile, we will further investigate the mentioned baseline and include it in the final version.
>
>  > "Can the authors provide more intuition on why they arrived at the proposed choice in Equation 6? Also, is there any other reason for choosing GCN apart from its simplicity and ease of incorporating the permutation equivariance?"
>
> The fundamental idea is that when using PE, one needs to emphasize the relative distance information between the nodes instead of the absolute positions of the nodes. This inspires us to arrive at Equation 6 because each entry of $ZZ^T$ essentially describes relative information between nodes. In this work, we choose GCN to do an instantiation of the proposed technique, because of its simplicity and ease of incorporating different types of equivariance.
>
> > "The zig-zag patterns in figure 4 are very interesting. I understand that it may or may not be trivial, but it will be helpful if the authors can provide some more intuition for such behavior apart from what is stated in section H.4."
>
> Due to the time limit before the submission due, we were only able to get each point of the experiments in Figure 4 by averaging over 2-time experiments. The updated plots by averaging over 10-time experiments are given in the revised manuscript.
>
> [1] https://en.wikipedia.org/wiki/Convolutional_neural_network
>
> [2] Pan Li and Jure Leskovec. “The expressive power of graph neural networks.” Graph Neural Networks: Foundations, Frontiers, and Applications, edited by Lingfei Wu, Peng Cui, Jian Pei, and Liang Zhao, Springer, 2021, pp. 63–98.
>
> [3] Mei, Song, Theodor Misiakiewicz, and Andrea Montanari. "Learning with invariances in random features and kernel models." COLT (2021).
>
> [4] S. Abu-El-Haija, B. Perozzi, R. Al-Rfou, and A. A. Alemi, “Watch your step: Learning node embeddings via graph attention,” Advances in Neural Information Processing Systems 31 (2018).

---

> > ### Author Response · Authors · 2021-11-23
> > **Response to Reviewer KJUw (Part II)**
> >
> > > "Can the authors elaborate the related works section 1.1 a bit more? Particularly, more details on the methods that propose an explicit use of PE techniques such as [5] and [6] might be helpful."
> >
> > Due to the page limit, we could only present the highlighted argument/contribution of each related work in Sec 1.1. We will attach a more detailed discussion of methods that explicitly use PE techniques including [5] and [6] in the Appendix of the final version.
> >
> > > "Although intuitively sound, as I have mentioned in the strengths subsection above, it will be great if the authors can provide some reference or more conclusive evidence to this statement - 'Inspired by the stability of the eigenspace, the idea to achieve PE-stability is to make the GNN layer invariant to the selection of bases of the eigenspace for the positional features' on the second last paragraph on page 5."
> >
> > The difference between Lemma 3.4 and Lemma 3.5 directly demonstrates the statement. For other references, we suggest that the reviewer check the proof of Lemma 3.5, where the Davis-Kahan theorem [7] and its variant [8] on eigenspace perturbation are adopted.
> >
> > [5] Srinivasan, Balasubramaniam, and Bruno Ribeiro. "On the equivalence between positional node embeddings and structural graph representations." arXiv preprint arXiv:1910.00452 (2019)
> >
> > [6] Dwivedi, Vijay Prakash, and Xavier Bresson. "A generalization of transformer networks to graphs." arXiv preprint arXiv:2012.09699 (2020).
> >
> > [7] Davis, Chandler, and William Morton Kahan. "The rotation of eigenvectors by a perturbation. III." SIAM Journal on Numerical Analysis 7.1 (1970): 1-46.
> >
> > [8] Yu, Yi, Tengyao Wang, and Richard J. Samworth. "A useful variant of the Davis–Kahan theorem for statisticians." Biometrika 102.2 (2015): 315-323.

---

### Official Review · Reviewer_L7CF · 2021-11-02

**Correctness:** 4
**Technical Novelty And Significance:** 3
**Empirical Novelty And Significance:** 3
**Recommendation:** 8
**Confidence:** 3

**Main Review:**

I appreciate the reasonable motivation and rigorous analysis in the paper. I only have some comments on it.

1) The analysis of matching is limited to graphs of the same size. Is there any possibility to generalize to different sizes? How does it consistent with experiments where two graphs are of different sizes?
2) The idea actually smartly borrows from SE(3)-transformer (or etc) with positional encoding (PE) to replace the physical coordinate. The inner product in eq (6) makes it only rotation equivariant while it is easy to use L2 distance (or etc) to make it also translation equivariant.
3) Do authors have any intuitive interpretation on PE, why do we need it to be rotation (or translation) equivariant? Some examples would help the audience further to understand the significance.

**Summary Of The Paper:**

The paper focuses on tasks of a set of nodes handled by graph neural networks (GNNs), which generally requires random feature (RF) or positional encoding (PE) to recognize the node identity. When RF is hard to converge and PE is less generalizable and stable to unseen graphs, authors propose a new architecture name PEG, processing node features and PE in different channels. Rigorous theoretical analysis shows PEG is permutation invariant to node features and rotation equivariant to PE, and its stability is guaranteed. Extensive experiments on link prediction (the task on node pairs) demonstrate the advantage of PEG.

**Summary Of The Review:**

I am generally satisfied with the content of the paper.

---

> ### Author Response · Authors · 2021-11-23
> **Response to Reviewer L7CF**
>
> We greatly appreciated Reviewer L7CF for recognizing our contributions and strongly supporting the acceptance of this work. Reviewer L7CF also insightfully observed the connection of our work to the SE(3)-transformer (or etc.). Actually, our idea was indeed inspired by that line of research. We respond to the two questions that reviewer L7CF raised.
>
> > "The analysis of matching is limited to graphs of the same size. Is there any possibility to generalize to different sizes? How is it consistent with experiments where two graphs are of different sizes?"
>
> Although GNNs are claimed to be able to generalize over graphs of different sizes, we think the rigorous characterization of that property or even just the formulation of that property is still missing in the community. The most relevant result, to the best of our knowledge, is in [1], while the theory is still far from complete. Indeed, rigorously formulating and characterizing such a problem is an interesting direction for us to look into in the future.
>
> > "Do authors have any intuitive interpretation on PE, why do we need it to be rotation (or translation) equivariant? Some examples would help the audience further to understand the significance."
>
> One approachable way to understand it is that positional encodings associate nodes in the graph with some absolute positions. However, there are no canonical absolute positions of the nodes in the graph. What really matters is the relative distance information between the nodes. This is also why the previous work that uses distance encoding [2] can naturally hold good generalization. To let the GNN model ignore the side-effects caused by the absolute positions of nodes while focusing on the relative positions of nodes, we need to let GNN hold such geometric equivariance properties.
>
> [1] Xu, Keyulu, et al. "How neural networks extrapolate: From feedforward to graph neural networks." arXiv preprint arXiv:2009.11848 (2020).
>
> [2] Li, Pan, et al. "Distance encoding: Design provably more powerful neural networks for graph representation learning.'' Advances in Neural Information Processing Systems 33 (2020).

---

### Official Review · Reviewer_rbLd · 2021-11-02

**Correctness:** 3
**Technical Novelty And Significance:** 3
**Empirical Novelty And Significance:** 2
**Recommendation:** 6
**Confidence:** 4

**Main Review:**

(a) -- Comments, with strengths:
- *Positioning*: The location of this work with respect to the literature is detailed and comprehensive as the limitations of existing works along this line is properly described. In particular, the discussion with Laplacian eigenvectors based PEs that could have ambiguities due to the sign and multiplicity is reviewed clearly and the work attempts to address the issues henceforth.
- *Writing*: The writing is easy to follow for readers aware with the research direction. Equivariance and stability of PEs to be used in GNNs is at the centre of the paper and their definitions, formalization and relevant conditions are clearly expressed.
- *Significance*: The contribution of the paper thus is helpful for the community given there are several papers recently that are attempting to use graph PEs to improve GNNs expressivity. In those works, the equivariance/stability issues of PEs were not explicitly addressed. The PEs were more like augmentations to GNNs in terms of supplying higher structural information when they learn on graphs; as nodes in graphs would not be canonically defined in terms of positions.
- *Empirical results/Experiments*: The focus in this work is only on link prediction tasks. Although I believe the work can be applied to general graph representation learning for other tasks as well. Experiments on the proposed model *compared to baselines* show the advantage of using the PE in the proposed model that uses these as edge weights. However, these models do not provide SOTA results and lag *behind the top scores* on the leaderboards of two OGB datasets considered [1]. In this respect, it may not be clear how the satisfiability of the two conditions leads to a powerful model, in practice.


(b) -- Other comments/concerns:
- In page 6, "We implement g in our model PEGN with further simplification". => Does this mean the paper does not present a GNN 'g' which is PE-equivariance and PE-stable in general? Are the results of satisfying the two conditions *tied* to the instantiation of g_PEG only? [Since, g_PEG seems a modification of the GCN with edge weights that are supplied with positional features of source and destination nodes.]
- In page 2, the paper claims "The key idea is to use separate channels to update the original node features and positional features." However, to the best of my understanding of the paper, I feel this is not (necessarily) enforced. For instance, the Eqn. 6 does *not update positional features*.
- In page 2, the paper states "PEG achieves comparable performance with strong baselines based on DE". I could not find comparisons on the datasets where PEG is compared against DE.
- During the discussion of non-uniqueness of LE as PE used in previous works, I believe simply using absolute signs of the eigenvectors would guarantee permutation equivariance; although I understand the stability issues.
- The title of the paper says ".... More Powerful Graph Neural Networks". I feel this addition of power is not discussed in the paper. I understand that its necessary to sort (and get rid of) the issues of Z_LE while using them as PE. But is the power being added just because of the equivariance/stability conditions met? This is also reflected in the experiments where the scores on the 2 OGB datasets do not compare to SOTA results. In other words, I am still unable to ascertain to myself whether the performance boost (compared to baselines) comes from i) simply the PE variants used (as they may be bringing other higher structural information to the GNN used). or ii) the addressal of the issues of equivariance and instability with the PEs? To this end, a fair study of two model versions where one version has the issues of instability and the other does not have, leaving all other settings intact for the two versions, would help addressing this question!



(c) Minor comments:
* Page 8; under the heading "Implementation details": typo constant feature (P) ==> constant feature (C)?
* The comparison of the results on the datasets with other GNN SOTA is not presented?
* In the experiment tables, the mean and s.d. is reported. How are these reported on? How many runs? These details can help reproducibility.



References:
[1] OGB link property prediction leaderboards: https://ogb.stanford.edu/docs/leader_linkprop/
[2] Srinivasan, B. and Ribeiro, B., 2019. On the equivalence between positional node embeddings and structural graph representations. arXiv preprint arXiv:1910.00452.



**Summary Of The Paper:**

The paper considers the topic of equivariant and stable positional encodings (PEs) for graph neural networks. It goes along a recent line of research that develop PEs for GNNs to: i) disambiguate structurally different nodes using positions and ii) improve the GNN in consideration.
This work formally studies the two conditions of equivariance and stability for graph PEs (Section 3.1). A new GNN layer is proposed (Section 3.2) based on modification of a GCN based layer with edge features that can satisfy the two conditions. Experiments are conducted on several small-scale link prediction datasets with performance improvements over baselines.


**Summary Of The Review:**

Overall, I feel this work is interesting and important in this direction of *designing PEs for GNNs* that do not have any issues violating the beauty of using GNNs in the first place; and critical for link prediction tasks as known from the literature [2]. However, it seems the empirical results may not be adequate to reflect on the technical contributions of the paper. In summary, my evaluation of ‘correctness’ and ‘technical novelty and significance’ measures of this paper are positive, while the evaluation of ‘empirical significance’ does not seem to align to the technical contribution.

---

> ### Author Response · Authors · 2021-11-23
> **Response to Reviewer rbLd (Part I)**
>
> We greatly thank Reviewer rbLd for appreciating our contributions to the theory and supporting the acceptance of this work. Reviewer rbLd’s concerns are mostly on the practical aspect, which we respond to as follows.
>
> > "Does this mean the paper does not present a GNN 'g' which is PE-equivariance and PE-stable in general? Are the results of satisfying
> the two conditions tied to the instantiation of g_PEG only?"
>
> To achieve PE-equivariance, we do not need a specific implementation of GNNs. PE-equivariance only depends on the two equivariant conditions: 1) Permutation equivariance w.r.t. all features; 2) Rotation equivariance w.r.t. positional features. We demonstrate this property in Theorem 3.6.
>
> PE-stability of course depends on the instantiation of the GNN layer because the parameter norms and the activation properties will both affect the stability coefficient. We prove it in Theorem 3.7. However, it does not mean that other implementations of $g$ are not PE-stable. For other forms of $g$, one can follow our proof strategy of Theorem 3.7 to check its stability.
>
> > "On page 2, the paper claims 'The key idea is to use separate channels to update the original node features and positional features.' However, to the best of my understanding of the paper, I feel this is not (necessarily) enforced."
>
> We claimed "The key idea is to use separate channels to update the original node features and positional features." because this is the condition asked by the theory. Theorem 3.6 (PE-equivariance) allows updating both features. Theorem 3.7 (PE-stability) can also be derived if both features get updated, though, in practice, it depends on the applications to decide whether one indeed needs to update both channels. We are happy to see that our theory inspires researchers to design other dedicated GNN layers for their applications besides PEG. Our theory lying upon updating both node features and positional features gives them more freedom to design. Of course, they need to satisfy the two equivariant properties highlighted by this work to guarantee good generalization and model stability. Here, we choose a rather simple implementation because it has given a good performance in the experiments over the datasets we use.
>
> > "[...] I could not find comparisons on the datasets where PEG is compared against DE."
>
> SEAL [1] adopts Double-Radius Node Labeling (DRNL) to compute deterministic distance features, which falls into the general idea of distance encoding [2]. PEG is compared against SEAL in both task 1 and task 2.
>
> > "During the discussion of non-uniqueness of LE as PE used in previous works, I believe simply using absolute signs of the eigenvectors would guarantee permutation equivariance; although I understand the stability issues."
>
> If there are multiple eigenvalues, adding absolute signs cannot ensure the uniqueness of PE and thus fails to be permutation equivariant. Please refer to Lemma 2.6.
>
> > "The model is not SOTA over the two OGB datasets."
>
> First, achieving the SOTA performance is never the goal of this work as we are to investigate the fundamental theories of GNNs and propose their principled practical inspiration. GCN is the most important baseline for this work because our implementation is built on GCN with some additional changes. This also gets appreciated by the other reviewers. Just based on the small change, our model has already achieved a good and uniform boost of performance. Therefore, we believe comparing PEG to GCN is sufficient to demonstrate the theory. To reach SOTA typically requires a lot of tricks, which loses the important theoretical insights provided by this work.
>
> That has been said, SEAL (one of the baselines) is almost the SOTA method for ogb-collab. Note that in the paper we compare the case without plugging in the validation links during the testing. Our method almost achieves the same performance as SEAL. The top-ranked one in the leaderboard for ogb-collab is a pure network embedding method, which can actually provide another type of PE that can be utilized by our model. By leveraging such PE, our model has the potential to achieve the SOTA. The SOTA method for ogb-ddi actually uses edge attributes, which can be further merged into our model. We leave the detailed fine-tuning of our model on different datasets for future investigation but we do not think they are necessary for this paper.
>
> [1] Zhang, Muhan, and Yixin Chen. "Link prediction based on graph neural networks." Advances in Neural Information Processing Systems 31 (2018): 5165-5175.
>
> [2] Li, Pan, et al. "Distance encoding: Design provably more powerful neural networks for graph representation learning.'' Advances in Neural Information Processing Systems 33 (2020).

---

> > ### Author Response · Authors · 2021-11-23
> > **Response to Reviewer rbLd (Part II)**
> >
> > > "I am still unable to ascertain to myself whether the performance boost (compared to baselines) comes from i) simply the PE variants used (as they may be bringing other higher structural information to the GNN used). or ii) the addressal of the issues of equivariance and instability with the PEs? To this end, a fair study of two model versions where one version has the issues of instability and the other does not have, leaving all other settings intact for the two versions, would help address this question!"
> >
> > Our experiments have already clarified the performance boost. By comparing our model with VGAE that uses node features + PE (deepwalk) or simply PE (deepwalk) in Tables 1 and 3, we demonstrate that the addressal of the issues of equivariance and instability achieves better generalization in both transductive and inductive tasks. Because VGAE using PE fails to be equivariant and stable. By comparing our model with VGAE that uses only node features in Tables 1 and 3, and GCN in Table 2, we demonstrate the importance of extra expressive power brought by the PEs.
> >
> > We further address this question by providing the study of PE generalization & stability experiments in the revised manuscript in Appendices I.1 and I.2.
> >
> > 1. In the task of Appendix I.1, stochastic block models (SBM) with 2 blocks are used to generate random graphs, and the probabilities of 0.3 and 0.1 are used to form the link between nodes within the block and across different blocks, respectively. Here we randomly select links inside the blocks as positive samples, and randomly select nonexistent links (unconnected node pairs) of the generated graph as negative samples. The model is trained by using various numbers of graphs (e.g. 1, 10, 30, 50), validated and tested on 10 random graphs respectively. For each graph that is used for training or testing, we utilize 10\% positive links and pair them with the same amount of negative missing links for training or testing respectively. We choose three variants of GAE that use constant features (degree), random features and positional features as baselines. All the models are trained until they converge and the models with the best validation performance are used to report the results, which are shown in Figure 5. For various sizes of input graphs, PEG achieves AUC around 0.98 by just using one training graph and slightly improves AUC by using more training graphs, which consistently outperforms all three GAE variants that even use the entire 50 training graphs with large margins. This directly demonstrates the stability and better generalization of PEG models.
> >
> > 2. In the task of Appendix I.2, we perturb the input graphs with a certain percentage of edges being randomly added or dropped before the inference, where the rest of the settings remain the same for link prediction. By comparing the model with different levels of input perturbation, we can observe from Figure 6 that PEG achieves consistently and significantly better performance with either node features or constant features in all four scenarios, which is far more robust than GAE/VGAE-based models that adopt PE under the same level of perturbations, particularly with heavy edge alterations. This result clearly shows that the performance boost indeed benefits from the equivariance and stability with PEs instead of simply using PE variants.
> >
> > > "In the experiment tables, the mean and s.d. is reported. How are these reported on? How many runs? These details can help reproducibility."
> >
> > The results in tables 1 and 2 are averaged over 10-time experiments, as claimed in the first paragraph of Sec. 4. We will further highlight in the final version. More detailed specifics to reproduce our experiments are given in Appendix H.

---

> > > ### Comment · Reviewer_rbLd · 2021-11-29
> > > **Response to authors**
> > >
> > > Thank you very much for your clarifications and the revised manuscript.
> > >
> > > I will retain the initial score in relation to the justification of 'More Powerful GNN'. I appreciate the hard work and efforts of the paper in adding additional experiments to show further generalization and applicability. However when we look at the two OGB datasets (-ddi and -collab) that are 'most meaningful' given the recent advance to evaluate GNNs on real-world larger datasets, the proposed architecture ($47.93$) does not beat simply GraphSage's performance ($53.90$) on -ddi, for instance. I believe a claimed 'Powerful GNN' would be expected to surpass simpler GNNs in such experiments.
> > >
> > > Nevertheless, I thank the authors for the paper's theoretical contribution which is useful for the community and I recommend with my initial score and status of acceptance.

---

> > > > ### Author Response · Authors · 2021-11-29
> > > > **Thanks!**
> > > >
> > > > Thank you so much for your suggestions and overall appreciation.
> > > >
> > > > In the follow-up work, we will work on using different backbones such as GraphSAGE with PE to check the performance (say for DDI) to demonstrate the universality of the benefit.

---

### Official Review · Reviewer_eyVz · 2021-11-04

**Correctness:** 3
**Technical Novelty And Significance:** 3
**Empirical Novelty And Significance:** 3
**Recommendation:** 6
**Confidence:** 4

**Main Review:**

Strengths:
1.	The proposed approach is well motivated and mathematically rigorous.
2.	The writing is in general clear and well-reasoned.
3.	The experimental results have demonstrated the superiority of the proposed approach on multiple datasets.


Weaknesses:
1.	The stability definition needs better justified, as the left side can be arbitrarily small under some construction of \tilde{g}. A more reasonable treatment is to make it also lower bounded.
2.	It is expected to see a variety of tasks beyond link predict where PE is important.


**Summary Of The Paper:**

This paper proposes a novel GNN layer called PEG layer to address the issue that existing positional encoding are not generalized to unseen graph very well. PEG designs different operators for raw features and positional features separately, which is able to ensure permutation equivariance and rotation equivariance for each type of feature, respectively. The experimental results have also demonstrated the power of the proposed approach.

**Summary Of The Review:**

This paper provides a through discussion on limitations of existing PE approaches and proposes the criterions of PE-equivariance and PE-stable. Mathematically, this paper proves the proposed PEG layer is able to preserve permutation equivariance as well as to achieve PE-stability. The proposed technique is solid and novel. There are some minor concerns as raised in weaknesses, but overall it is a paper worth of accept.

---

> ### Author Response · Authors · 2021-11-23
> **Response to Reviewer eyVz**
>
> We greatly thank Reviewer eyVz for appreciating our contributions to the theory and supporting the acceptance of this work. Here, we are to respond to the two weaknesses proposed by Reviewer eyVz.
>
> > "The stability definition needs to be better justified, as the left side can be arbitrarily small under some construction of \tilde{g}. A more reasonable treatment is to make it also lower bound."
>
> We are not sure about what exact form of the lower bound Reviewer eyVz is looking for. We guess that Reviewer eyVz might refer to the lower bound $c*d(\mathcal{G}^{(1)}, \mathcal{G}^{(2)})$ for some positive constant $c$, which essentially means the change of the GNN output is almost isotropic/equivalent with respect to the change of the input by being paired with the upper bound. Actually, the definition of model stability does not require such a lower bound [1]. Adding a lower bound to satisfy the isotropic/equivalent requirement is stronger, but we do not think it is necessary in general graph learning models.
> 1. First, it is unnecessary for most graph learning tasks. For example, if we are to classify the nodes, node-pairs, or the whole graphs, the final goal of the model is to classify the input nodes/node-pairs/graphs into several categories. The inputs that are put in the same category actually share the same outputs of the model (and thus the distance between the model outputs is 0, 0 on the LHS of Eq. (1)), although they may have different input graph structures. Therefore, the lower bound is unnecessary here to make a good prediction.
> 2. Second, the lower bound is somehow related to another notion, the GNN expressive power, though they are not the exact same. The expressive power means if the inputs are different (under any permutation), GNNs may be able to distinguish them (associate them with different output). In the study of GNN expressive power, the goal is to guarantee that there exist some parameters with which GNNs may distinguish different inputs rather than the "always needed" lower bound asked by the reviewer, which indicates that GNNs always have to associate different inputs with different outputs.
>
> > "It is expected to see a variety of tasks beyond link predict where PE is important."
>
> We provide a preliminary experiment regarding the node classification task on citation networks -- Cora, Citeseer, and PubMed.
>
> We closely follow the transductive experimental settings in [2]. The results of our evaluation are summarized in Appendix I.3. Both PEG-LE and PEG-DW significantly outperform GCN and provide comparable results against GAT, which illustrates the effectiveness of the proposed PEG on node-level tasks. Note that we almost did not tune our model to achieve better results. We believe even better results may be achieved by finer tuning. Moreover, GAT uses multi-head attention to aggregate information from neighbors where the attention is based on node features. In comparison, PEG adopts simple GCN layers where the edge weights are based on positional encodings that only depend on the network structure. The information sources are different. Hence, we believe PEG has the potential to be combined with the attention mechanism based on node features to further get better predictions, though more effort on model tuning is needed.
>
> [1] https://en.wikipedia.org/wiki/Stability_theory
>
> [2] Yang, Zhilin, William Cohen, and Ruslan Salakhudinov. "Revisiting semi-supervised learning with graph embeddings." International Conference on Machine Learning. PMLR, 2016.

---

> ### Author Response · Authors · 2021-11-29
> **We are looking forward to your comments**
>
> We thank reviewer eyVz again for the pointed comments.
>
> We were wondering if the additional experiments provide the information you were looking for. We are looking forward to your further comments.

---

### Official Review · Reviewer_FBSc · 2021-11-04

**Correctness:** 1
**Technical Novelty And Significance:** 3
**Empirical Novelty And Significance:** 1
**Recommendation:** 6
**Confidence:** 5

**Main Review:**

Strengths: the paper is well written and I liked the theoretical part (Section 3). The ideas are very natural and nicely presented. The notion of stability will likely be picked up in future works and the authors show how to construct PE-stable layers which can be useful.

Weakness: the evaluation in Section 4 is very disappointing. Equivariance is only useful for inductive task but the link prediction task considered by the authors is a transductive task. For task 1, I do not see why you would ask your algorithm to be equivariant as the learning algorithm is trained on one graph. In such a case, you can choose an ID for each node and use this ID to make your link prediction. Task 2 is better to test equivariant architecture. But for task 2, I do not understand why the authors look at the link prediction problem. For example, the PPI dataset is typically not used to do link prediction so why doing this here?
Indeed, the authors are aware of this weakness as they write: 'Both tasks may reflect the effectiveness of a model while Task 2 may better demonstrate the model’s generalization capability that strongly depends on permutation equivariance and stability.'
To validate their theoretical results, the authors should consider a task where equivariance is 'crucial'. The QAP problem studied in 'Expressive Power of Invariant and Equivariant Graph Neural Networks' by Azizian and Lelarge (ICLR 2021) is probably a good candidate.


**Summary Of The Paper:**

This paper studies positional encoding (PE) for Graph Neural Networks (GNN). A positional encoding can be seen as a computed feature for each node from characteristics of the graph. A well-known example of PE are the components of the eigenvectors of the Laplacian of the graph. Unfortunately, such a PE is not permutation equivariant. This is mainly due to the fact that eigenvectors are not uniquely defined.
This paper address this issue with some care (correcting some mistakes made in previous works).
Section 3 of the paper presents the main theoretical results of the paper. In particular, the authors introduce a notion of stability that is stronger than equivariance: if two input graphs are 'similar' then their outputs through a stable layer should also be 'similar' with a 'Lipschitz' constant. This notion is interesting. Then, the authors explain some errors made in the literature and address these issue by proposing a PE-stable layer that is shown to be PE-stable under some conditions on the spectrum of the Laplacian. Some other techniques to get PE-stable layers are provided in section 3.3
In Section 4, experimental results are provided with this new architecture for task 1: link prediction and task 2: domain-shift link prediction. These experiments demonstrate the effectiveness of PE-stable layers.

**Summary Of The Review:**

There is a mismatch between the theoretical part of the paper and the experiments.

---

> ### Author Response · Authors · 2021-11-23
> **Response to Reviewer FBSc (Part I)**
>
> We greatly thank Reviewer FBSc for appreciating our contributions to the theory. We believe the new notion of stability is crucial for the entire community to better understand the relationship between generalization and the equivariance/invariance property of GNNs. However, we respectfully disagree with Reviewer FBSc’s argument about the mismatch between our theory and the experiments.
>
> > "Equivariance is only useful for the inductive task but the link prediction task considered by the authors is a transductive task."
>
> * This statement is incorrect.
>
> 1. First, equivariant and generalizable models are definitely useful for transductive tasks. Even if the model training and testing are conducted over the same graph, equivariant models are crucial to generalize the learned patterns across different sections of the graph for accurate predictions. For a toy example in Fig. 1, suppose (b,c) and (c,d)’s future link labels are known. Then, an equivariant GNN is able to generalize the link pattern learnt from (b,c) to (a,d), and from (c,d) to (a,b). However, the idea by using node IDs proposed by Reviewer FBSc will fail here.
>
> 2. Actually, a lot of previous works have leveraged the equivariant property to achieve better performance even for transductive tasks. For example, all the experiments in [1] (including node, node-pair, node-triad tasks) were conducted by performing model training and testing over the same graphs, while the model generalization and equivariance have brought great benefit in the model performance. Note that the invariance over different nodes, node-pairs, node-triads emphasized by [1] is equivalent to the equivariance over the whole graphs. Also, many models that emphasize invariance particularly for link-level tasks, such as [2][3][4][5], have also been evaluated in the transductive setting and can easily outperform other baseline models. This again demonstrates the effectiveness and importance of equivariance and generalization even in the transductive setting.
>
> 3. The work [6] that started investigating the combination of GNNs and the positional information was entirely built upon link-level tasks. [6] also considered both transductive and inductive settings. We choose to use link prediction by following the convention used in [6]. Link prediction is an important task to demonstrate whether the model may well capture and utilize the positional information of nodes in the graph. Note that Reviewers KJUw, L7CF, and rbld all agree on the importance to evaluate link prediction in our setting. Particularly, Reviewer KJUw found our domain-shift link prediction task (task 2) are practical and highly interesting.
>
> >  "In such a case, you can choose an ID for each node and use this ID to make your link prediction."
>
> * Using ID for each node may yield model instability.
>
> Besides the failure example in Fig.1 discussed in the above response, Reviewer FBSc can also understand why equivariance is important for transductive tasks by considering the proposed node ID idea more carefully. If node IDs are adopted, the model learns every node's personalized behavior. In this sense, the prediction will be very sensitive to even tiny graph perturbation: If we slightly change one node’s attributes or its contextual structure, the prediction for it may get substantially changed because no knowledge/patterns from other nodes can be inductively learned and referred to. In contrast, keeping model equivariance can easily avoid such an unstable issue.
>
> [1] Li, Pan, et al. "Distance encoding: Design provably more powerful neural networks for graph representation learning.'' Advances in Neural Information Processing Systems 33 (2020).
>
> [2] Zhang, Muhan, and Yixin Chen. "Link prediction based on graph neural networks." Advances in Neural Information Processing Systems 31 (2018): 5165-5175.
>
> [3] Teru, Komal, Etienne Denis, and Will Hamilton. "Inductive relation prediction by subgraph reasoning." International Conference on Machine Learning. PMLR, 2020.
>
> [4] Zhu, Zhaocheng, et al. "Neural Bellman-Ford Networks: A General Graph Neural Network Framework for Link Prediction." Advances in Neural Information Processing Systems 34 (2021).
>
> [5] Zhang, Muhan, and Yixin Chen. "Inductive matrix completion based on graph neural networks." ICLR. 2020.
>
> [6] You, Jiaxuan, Rex Ying, and Jure Leskovec. "Position-aware graph neural networks." International Conference on Machine Learning. PMLR, 2019.

---

> > ### Author Response · Authors · 2021-11-23
> > **Response to Reviewer FBSc (Part II)**
> >
> > > "Task 2 is better to test equivariant architecture. But for task 2, I do not understand why the authors look at the link prediction problem. For example, the PPI dataset is typically not used to do link prediction so why do this here?"
> > * PPI prediction is a very significant application.
> >
> > Because of the importance of task 1, we also disagree with Reviewer FBSc’s statement that task 2 is just to complement the “weakness” of task 1. Task 2 is actually used to further demonstrate the generalization capability of the models. Also, we do not see any issue with using the PPI dataset for link prediction. Reviewer FBSc’s reason that argues against using PPI for link prediction seems to be that previous works have used PPI for node classification, which is unwarranted and irrelevant. Actually, link prediction on the PPI dataset is practically significant as it is crucial to complement the missing information in biological experiments [7]. Many previous works indeed investigate the link prediction task on the PPI dataset. For example, searching Google Scholar with “PPI link prediction” reveals 29,300 results. The famous survey paper for link prediction [8] uses PPI as a benchmark dataset. The dataset PPA used for link prediction on open graph benchmark [9] is also a PPI network.
> >
> > > "The QAP problem studied in Azizian and Lelarge (ICLR 2021) is probably a good candidate".
> > * We did new experiments to further evaluate the model generalization and stability in the revised manuscript.
> >
> > Many thanks for the suggested experiments. We will investigate the QAP problem and include our findings in the final version. Due to the limited time for preparing the response, we focus on more realistic and practical tasks such as link prediction and node classification that other reviewers are also interested in.
> >
> > To further demonstrate the generalization, we conduct inductive link prediction over random graph models in Appendix I.1. For this task, stochastic block models (SBM) with 2 blocks are used to generate random graphs, and the probabilities of 0.3 and 0.1 are used to form the link between nodes within the block and across different blocks, respectively. Here we randomly select links inside the blocks as positive samples and randomly select nonexistent links (unconnected node pairs) of the generated graph as negative samples. The model is trained by using various numbers of graphs (e.g. 1, 10, 30, 50), validated, and tested on 10 random graphs respectively. For each graph that is used for training or testing, we utilize 10\% positive links and pair them with the same amount of negative missing links for training or testing respectively. We choose three variants of GAE that use constant features (degree), random features, and positional features as baselines. All the models are trained until they converge and the models with the best validation performance are used to report the results, which are shown in Figure 5. For various sizes of input graphs, PEG achieves AUC around 0.98 by just using one training graph and slightly improves AUC by using more training graphs, which consistently outperforms all three GAE variants that even use the entire 50 training graphs with large margins. This directly demonstrates the stability and better generalization of PEG models.
> >
> > [7] Lei, Chengwei, and Jianhua Ruan. "A novel link prediction algorithm for reconstructing protein–protein interaction networks by topological similarity." Bioinformatics 29.3 (2013): 355-364.
> >
> > [8] Lü, Linyuan, and Tao Zhou. "Link prediction in complex networks: A survey." Physica A: statistical mechanics and its applications 390.6 (2011): 1150-1170.
> >
> > [9] Hu, Weihua, et al. "Open graph benchmark: Datasets for machine learning on graphs." arXiv preprint arXiv:2005.00687 (2020).

---

> > > ### Comment · Reviewer_FBSc · 2021-11-30
> > > **thank you for your answer**
> > >
> > > I agree with you that my claim about equivariance and transductive task is not correct as equivariance is likely to help generalization. But I would still have liked to see a task where equivariance is 'more crucial'.

---

> > > > ### Author Response · Authors · 2021-12-01
> > > > **Thank you for increasing your evaluation**
> > > >
> > > > Thank you for increasing your evaluation! We take your suggestion seriously. We are working on a follow-up work that focuses on those tasks with multiple graphs where equivariance is more crucial as you suggested.

---

> ### Author Response · Authors · 2021-11-29
> **We are looking forward to your comments**
>
> We thank reviewer FBSc again for the pointed comments.
>
> We were wondering if we have addressed your concerns on the notion of generalization for the link prediction task and the experiments, whether you are satisfied with the next experiments. We are looking forward to your further comments.

---

### Author Response · Authors · 2021-11-23
**Overview of the response**

* We have clarified the misunderstanding with regard to the mismatch between theory and the experiments, as a response to Reviewer FBSc.

* We have added three experiments of link prediction on random graphs, perturbed graphs, and node classification tasks to demonstrate the generalizability (for Reviewers FBSc, rbLd, KJUw), stability (for Reviewer rbLd), and wide applicability (for Reviewer eyVz, KJUw) of our proposed PEG in Appendix I.

* We have fixed typos and adjusted some content to better improve the readability.

We thank all reviewers for their valuable feedback and comments. We look forward to your response.

---

### Decision · Program_Chairs · 2022-01-20

**Decision:**

Accept (Poster)

**Comment:**

This work studies the question of increasing the expressive power of GNNs by adding positional encodings while preserving equivariance and stability to graph perturbations.
Reviewers were generally positive about this work, highlighting its judicious problem setup, identifying the right notion of stability and how it should drive the design of positional encodings. Despite some concerns about the discrepancy between the theoretical results and the empirical evaluation, the consensus was ultimately that this work is an interesting contribution, and therefore the AC recommends acceptance.